# SpecExit: Accelerating Large Reasoning Model via Speculative Exit

## Abstract

Despite their strong performance on reasoning tasks, large reasoning models (LRMs) often suffer from overthinking, producing unnecessarily long outputs and incurring high end-to-end latency, a significant limitation to their real-world deployment. To address overthinking, early-exit mechanisms have been proposed to terminate reasoning before typical completion, showing that this approach can effectively shorten generation length with minimal impact on accuracy. However, their reliance on probing mechanisms introduces a detection overhead that limits their end-to-end latency gains and compromises their generalizability across diverse problems. Inspired by the use of hidden states in speculative decoding, we propose **SpecExit**, a novel framework that predicts both future tokens and an early-exit signal directly from a lightweight draft model without probing overhead. Our method offers significant improvements, achieving up to 66% generation length reduction and 2.5× end-to-end speedup compared with the speculative decoding baseline, without compromising accuracy. Our method leverages the inherent signals from hidden states to provide effective early-exit signals, suggesting broader use of hidden states for efficient reasoning. Our code is available at: `https://anonymous.4open.science/r/SpecExit-B802`.

## 1 Introduction

Large reasoning models (LRMs) such as OpenAI-o1 (OpenAI, 2024), DeepSeek-R1 (DeepSeek-AI et al., 2025) and Qwen (Qwen et al., 2025) have recently achieved state-of-the-art performance in complex tasks. These models follow the test-time scaling law (Snell et al.; Brown et al.; Muennighoff et al.), where generating longer chain-of-thought (CoT) sequences (Wei et al.) generally enhances model performance. However, this reliance on extended reasoning often leads to an *overthinking* problem, where models produce unnecessarily verbose outputs. This redundancy leads to both excessive token usage and high end-to-end latency, which limits LRMs' practical deployment.

To mitigate overthinking, researchers have proposed both inference-time and training-based strategies (Sui et al., 2025). Inference-time early-exit methods (Yang et al., 2025; Fu et al., 2024) rely on model-generated signals such as intermediate answers or output logits to terminate decoding once sufficient evidence is detected. These methods can shorten reasoning length without harming accuracy, but the probing overhead they introduce limits the actual latency gains. Training-based approaches, such as reinforcement learning (Aggarwal & Welleck, 2025; Yeo et al., 2025) and supervised fine-tuning (Ma et al., 2025b; Munkhbat et al., 2025), incur little runtime overhead during inference but risk altering the model's output distribution. As a result, existing methods struggle to deliver consistent improvements in end-to-end efficiency.

Speculative decoding (Chen et al., 2023; Leviathan et al., 2023) is a promising approach that improves efficiency without altering the target model's outputs. A lightweight draft model proposes multiple candidate tokens in advance, and the target model then verifies these candidates in parallel. This parallel verification allows the system to generate multiple tokens per forward pass of the target model, alleviating the inherent sequential bottleneck of autoregressive decoding and better utilizing modern GPU hardware. However, this strategy alone does not resolve the overthinking problem, as models still generate the full CoT. Recent advanced speculative decoding methods (Li et al., 2024a; Zhang et al., b) exploit hidden states to predict several future tokens. Other studies (Lin et al.; Dong et al.) also show that hidden states encode richer predictive signals beyond next-token probabilities.

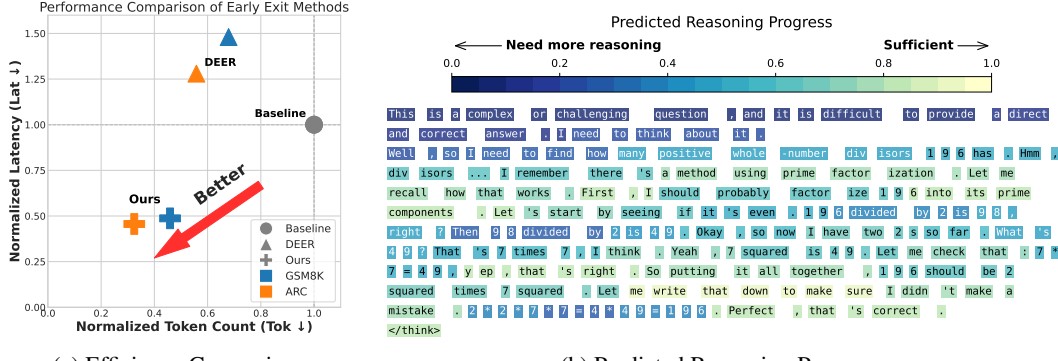

(a) Efficiency Comparison        (b) Predicted Reasoning Progress.

Figure 1: Effectiveness of the proposed method. (a) Statistical comparison showing that our approach produces shorter reasoning chains and faster inference than baselines. (b) visualizes the predicted reasoning progress on a MATH500 example, where darker colors denote insufficient reasoning and lighter colors denote sufficiency, demonstrating valuable signals can be extracted from hidden states regarding the model's reasoning process.

In this work, we introduce **SpecExit**, a reasoning-aware early-exit framework that leverages draft model hidden states not only to anticipate future tokens but also to predict early-exit signals. Unlike prior probing-based approaches, SpecExit requires no modifications to the target model and incurs no additional detection overhead. Instead, it extends the lightweight draft model with auxiliary prediction heads, enabling it to jointly output token distributions and reasoning-related signals in a single forward pass. By exploiting the latent information embedded in hidden states, SpecExit provides reliable criteria for dynamically terminating chain-of-thought generation when sufficient reasoning has been achieved.

We validate SpecExit on state-of-the-art reasoning models across mathematical, scientific, and logical benchmarks. For Qwen3-4B-Thinking-2507, DeepSeek-R1-Distill-Llama-8B, and Phi-4-reasoning models, SpecExit achieves up to 66% generation length reduction and 2.5× end-to-end latency speedup compared with the speculative decoding baseline. Our contributions can be summarized as follows:

- **Signals Extracted for Early Exit**. We derive early-exit signals from hidden features and integrate them into speculative decoding, enabling reliable early exit for efficient reasoning.

- **General and Practical Framework**. We implement SpecExit, a reasoning-aware early-exit framework, in both PyTorch and vLLM, making it easy to deploy across diverse inference environments.

- **Substantial End-to-End Performance Gains**. SpecExit reduces reasoning length by as much as 66% and delivers as high as 2.5x speedup in end-to-end inference compared with speculative decoding while maintaining accuracy.

## 2 MOTIVATION

Current early-exit methods face two challenges: runtime overhead from probing and limited generality from task-specific prompts. Since hidden states already reflect reasoning sufficiency, we propose leveraging them to replace costly probing for faster and more reliable inference across diverse tasks.

**Probing Overhead and Limited Generalizability.** Probing-based early-exit methods suffer from both latency overhead and poor generalizability across tasks. Shortened reasoning traces do not effectively translate into latency improvements. As shown in Figure 1a, DEER (Yang et al., 2025) reduces the generation length of GSM8K (Cobbe et al., 2021) and ARC-Challenge(Clark et al., 2018) by 32% and 44% respectively on Qwen3-4B-Thinking-2507, but even increases end-to-end latency than vanilla baseline. The gap arises because probing introduces extra computation, and its effectiveness is highly sensitive to both the task and the model. A probing phrase like "Final Answer is" elicits effective intermediate answers in math problems, but fails in coding tasks where

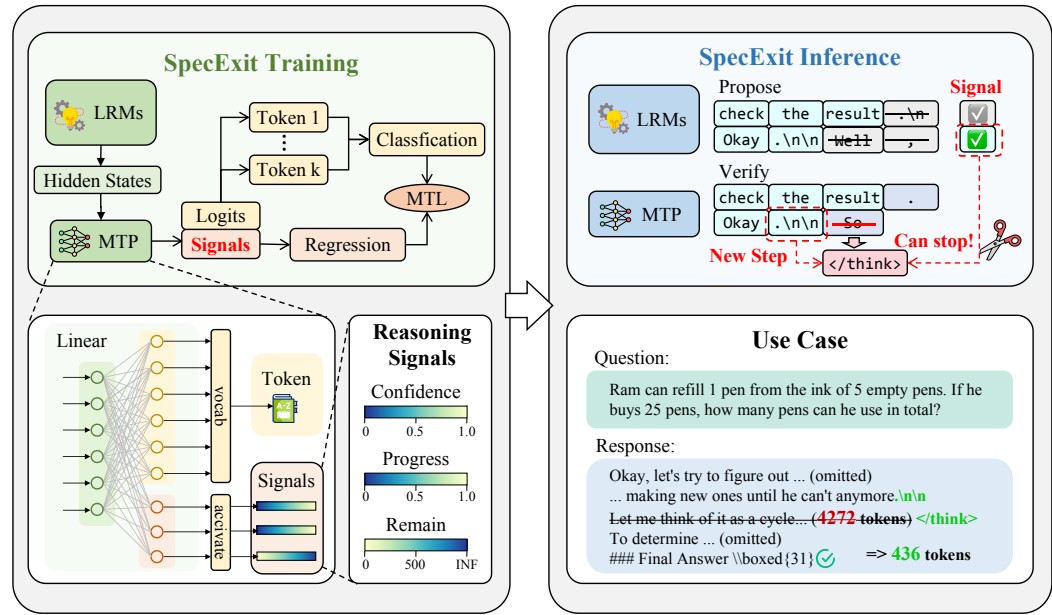

Figure 2: Overall architecture of the proposed SpecExit framework. The Multi-Token Prediction (MTP) layer is augmented to output both token logits and auxiliary signals. Training is performed with Multi-Task Learning (MTL), while at inference these signals guide speculative early stopping without modifying the backbone model. The example illustrates how redundant reasoning steps can be pruned while preserving final answer quality.

extra tokens are still generated. This drawback not only undermines generalizability across domains but also limits true latency savings.

**Signals from Hidden States**. Our preliminary experiments show that models' hidden states encode informative signals about its reasoning process. As illustrated in Figure 1b, we use a MLP trained on hidden states to predict reasoning progress. For complex tasks, the predicted signals appear darker at the beginning, reflecting the need for continued reasoning, but gradually shift to lighter colors as the model approaches a sufficient chain of thought. This progression suggests that hidden states provide fine-grained indicators of task complexity and reasoning sufficiency. Leveraging these internal signals offers an efficient alternative to costly probing, motivating our approach of utilizing signals from hidden states.

## 3 METHOD

### 3.1 OVERALL ARCHITECTURE

**SpecExit Framework**. The overall design of this work aims to incorporate additional learnable signals into large model reasoning, thereby providing explicit decision-making criteria for early stopping. In the decoding process of large language models, hidden states not only encode semantic and contextual information for next-token prediction but also implicitly contain higher-level cues related to reasoning progress, generation quality, and content completeness. The Multi-Token Prediction (MTP) mechanism leverages these hidden states to project into the vocabulary space and predict multiple future tokens simultaneously, thereby improving inference efficiency. Inspired by this, we extend the MTP layer while keeping the backbone language model unchanged, introducing auxiliary prediction heads that allow the model to explicitly generate reasoning-related signals, including confidence, reasoning progress, and remaining reasoning length, alongside token distributions. This design preserves the original language modeling ability while providing learnable auxiliary variables for inference control, enabling efficient and dynamic reasoning regulation. The overall architecture of the SpecExit framework is shown in Figure 2.

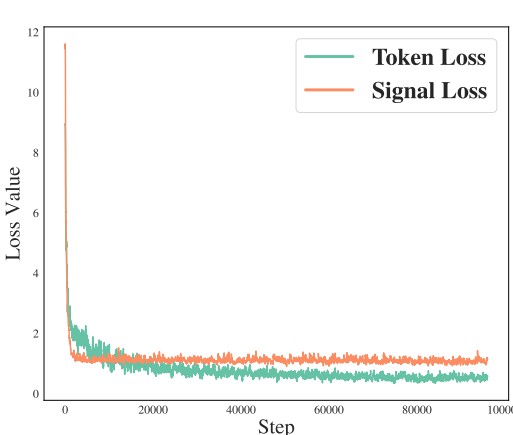

Figure 3: Convergence of token classification loss and signal regression loss during training.

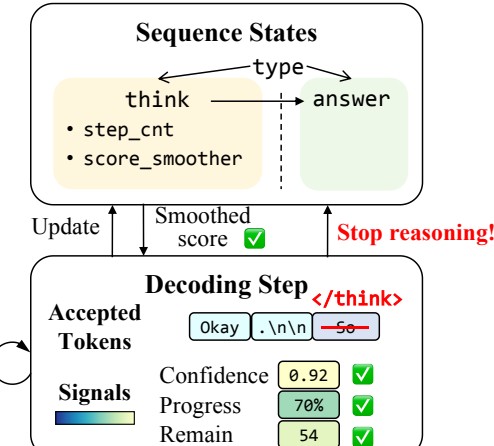

Figure 4: Inference process with signal-guided speculative exit.

**Model Structure**. We propose a speculative sampling-based early stopping mechanism for reasoning. In the MTP layer of the model, we extend the linear projection to include additional dimensions. In standard decoding, the hidden state $h \in \mathbb{R}^D$ is projected into the vocabulary space to predict the next token distribution, where $D$ is the hidden size of the model. In our method, the linear layer output is extended as:

$$Wh = \begin{bmatrix} W_{\text{tok}}h, \ W_{\text{conf}}h, \ W_{\text{prog}}h, \ W_{\text{rem}}h \end{bmatrix}, \tag{1}$$

where $W_{\text{tok}}h$ produces standard token predictions, while $W_{\text{conf}}h, W_{\text{prog}}h, W_{\text{rem}}h$ predict confidence, reasoning progress, and remaining reasoning length, respectively. These additional signals serve as observable indicators for deciding whether to stop reasoning early, thus reducing redundant computation without compromising output quality.

## 3.2 SIGNAL-EXTRACTED TRAINING

**Data Construction**. We first obtain the complete response generated by the base language model and extract the reasoning content enclosed within the `<think>` and `</think>` tokens. To identify the effective reasoning trace, we iteratively attempt to insert the closing marker `</think>` after each paragraph and verify whether the resulting final answer matches the original output. If the answer remains consistent, the subsequent reasoning content is regarded as redundant. Consequently, only the minimal reasoning segment required to produce the correct answer is retained as training data.

**Signal Annotation**. CONFIDENCE is defined as the geometric mean of the logit probabilities across prediction steps, reflecting the reliability of the generation; REMAINING reasoning length is defined as the number of tokens from the initial `<think>` marker to the earliest valid insertion point of `</think>` that still yields the correct answer; PROGRESS is represented as a normalized value increasing from 0 to 1, capturing the relative progression of the reasoning CoT.

**Signal Regression**. We propose a cost-efficient extension by introducing a small number of additional dimensions into the linear projection layer of the MTP module for regressing reasoning signals. These dimensions are orthogonal to the vocabulary classification weights, ensuring that signal regression does not interfere with the convergence of speculative decoding training.

The Multi-Task Learning (MTL) overall training objective jointly optimizes token classification and signal regression, defined as:

$$\mathcal{L} = \mathcal{L}_{\text{cls}} + \lambda_c \mathcal{L}_{\text{conf}} + \lambda_p \mathcal{L}_{\text{prog}} + \lambda_r \mathcal{L}_{\text{rem}}, \tag{2}$$

where $\mathcal{L}_{\text{cls}}$ is the standard cross-entropy loss for vocabulary prediction, and $\mathcal{L}_{\text{conf}}, \mathcal{L}_{\text{prog}}, \mathcal{L}_{\text{rem}}$ correspond to the regression losses for confidence, progress, and remaining reasoning length, with $\lambda_c, \lambda_p, \lambda_r$ denoting dynamic weighting coefficients. Specifically:

Confidence and progress are optimized using mean squared error (MSE), remaining reasoning length is optimized with mean squared logarithmic error (MSLE):

$$\mathcal{L}_{\text{conf}} = \frac{1}{N} \sum_{i=1}^{N} (\text{sigmoid}(\hat{c}_i) - c_i)^2, \tag{3}$$

$$\mathcal{L}_{\text{prog}} = \frac{1}{N} \sum_{i=1}^{N} (\text{sigmoid}(\hat{p}_i) - p_i)^2, \tag{4}$$

$$\mathcal{L}_{\text{rem}} = \frac{1}{N} \sum_{i=1}^{N} (\hat{r}_i - \log(1 + r_i))^2, \tag{5}$$

where $\hat{c}_i$, $\hat{p}_i$, and $\hat{r}_i$ represent the model-predicted confidence, progress, and remaining reasoning length, respectively; $c_i$, $p_i$, and $r_i$ denote their corresponding ground-truth values; and $N$ is the total number of samples.

**Dynamic Weighting**. Since the regression losses of signals converge faster than the token classification loss, we adopt a gradient-based dynamic weighting strategy to balance the contributions of different tasks. This mechanism assigns higher weights to tasks with smaller gradient magnitudes, preventing tasks with larger gradients from dominating the learning process and ensuring all tasks are effectively optimized. Formally, the mechanism is defined as:

$$\lambda_j = \frac{\|\nabla_\theta \mathcal{L}_j\|}{\sum_k \|\nabla_\theta \mathcal{L}_k\|}, \quad \mathcal{L}_{\text{total}} = \mathcal{L}_{\text{cls}} + \sum_j \lambda_j \mathcal{L}_j, \tag{6}$$

where $\mathcal{L}_j \in \{\mathcal{L}_{\text{conf}}, \mathcal{L}_{\text{prog}}, \mathcal{L}_{\text{rem}}\}$, $\nabla_\theta \mathcal{L}_j$ denotes the gradient of task $j$ with respect to the model parameters, $\mathcal{L}_{\text{cls}}$ is the cross-entropy loss for token classification, and $\lambda_j$ is the dynamically computed weight. This formulation ensures that the gradient contributions of all tasks are balanced, which facilitates stable convergence in multi-task optimization, as shown in Figure 3.

## 3.3 SIGNAL-GUIDED INFERENCE

**Overall Procedure**. We build upon the speculative decoding framework, where a smaller draft model first proposes a sequence of candidate tokens, which are then verified in parallel by a larger target model. To evaluate feasibility and efficiency, the inference procedure is implemented on both **PyTorch** and **vLLM** frameworks. The central modification lies in the forward pass of the target model. Beyond computing the logits for the next token, we additionally extract the final hidden state corresponding to the last accepted token. This representation is processed through a lightweight linear layer to generate three signals: a confidence score, a progress indicator, and an estimate of the remaining reasoning length, as shown in Figure 4.

**Speculative Decoding Inference Procedure.** In the speculative decoding inference pipeline, each sequence of draft tokens proposed by the draft model is forwarded to the target model for verification. Only those draft tokens that pass this verification are directly committed to the KV-cache of the target model, and the last accepted draft token is followed by a recover token generated by the target model. The hidden states produced by the target model during verification are then fed back into the draft model to guide the generation of new draft tokens. Subsequently, both the recover token and the newly generated draft tokens are again passed to the target model for verification and potential acceptance. A schematic overview of the speculative decoding pipeline integrated with the proposed reasoning early-exit mechanism is provided in Figure 11(Appendix).

**Stopping Conditions**. To ensure that early-exit decisions occur at semantically coherent boundaries, we introduce a class of special markers called *step split tokens*, which indicate natural segmentation

---

**Algorithm 1:** Inference procedure with signal-guided speculative exit

---

**Input:** Draft model $M_d$, target model $M_t$, tokenizer, thresholds
**Output:** Generated sequence $y$
Define $is\_thinking \leftarrow$ true;
Define $step\_split\_tokens \leftarrow \{$ids of "\n\n", ". \n\n", ...$\}$;
Define $stop\_think\_token \leftarrow$ id of </think>;
**while** *not terminated* **do**
    Extract hidden state of $t_{acpt}$, generate candidate tokens with $M_d$;
    Compute $signals$ (confidence, progress, remaining);
    Set $signals \leftarrow$ update smoothed scores;
    Concat last accepted token with draft candidates, forward through $M_t$ with tree attention;
    Accept tokens $t_{acpt}$, accept length $l_{acpt}$, target model recover token $t_{rec}$ ;
    **if** *$is\_thinking$ **and** $any(t_{acpt} \in step\_split\_tokens)$ **and** signals exceed thresholds* **then**
        Set $l_{acpt} \leftarrow$ corresponding $step\_split\_token$ position;
        Set $t_{rec} \leftarrow stop\_think\_token$;
        Update KV-cache and hidden states accordingly;
        Set $is\_thinking \leftarrow$ false;
    **end**
**end**

---

points in the generated text. Specifically, step split tokens can be divided into two categories: PARAGRAPH DELIMITERS (e.g., . \n\n), which mark the end of a paragraph or reasoning unit, and DISCOURSE MARKERS (e.g., "Wait", "But", or "Therefore"), which often signal semantic transitions or logical shifts during reasoning. Since the segmentation strategy based on PARAGRAPH DELIMITERS is more general, this strategy is adopted by default in subsequent experiments. Examples of commonly observed discourse markers in reasoning traces are shown in Figure 12 (Appendix). When a sampled token belongs to the above set, the early-exit logic is triggered. If the smoothed signal exceeds the predefined threshold, the system determines that the reasoning process has been sufficiently explored. The complete inference process is summarized in Algorithm 1. In this case, the accepted output length is truncated at the position of the step split token, and the target model's recover token is replaced with a special reasoning-end marker (e.g., </think>), thereby ensuring that the termination point lies at a natural boundary while maintaining coherence of the generated text.

**Signal Smoothing**. Since raw signals may exhibit significant volatility, relying directly on them risks premature or unstable termination. To enhance robustness, we apply an Exponentially Weighted Moving Average (EWMA) to smooth the signals across steps. At each iteration, the smoothed value is updated as a weighted average of the current raw signal and the previous smoothed value, with the smoothing factor controlling the balance between recent and past observations. A smaller factor emphasizes historical stability, yielding smoother traces that are less sensitive to transient noise. This ensures that termination decisions reflect consistent trends rather than isolated fluctuations.

## 4 EXPERIMENTS

### 4.1 EXPERIMENTAL SETUP

To evaluate the effectiveness of our SpecExit framework, we conducted a comprehensive set of experiments across multiple domains. Specifically, we used the **GSM8K** (Cobbe et al., 2021), **MATH500** (Hendrycks et al., 2021) and **AIME** (MAA Committees) datasets for mathematical reasoning, the **HumanEvalPlus** (Liu et al., 2023) dataset for coding, the **GPQA Diamond** (Rein et al., 2023) dataset for science, and the **ARC-Challenge** (Clark et al., 2018) dataset for logic. Experiments are conducted on three mainstream LRMs: **Qwen3-4B-Thinking-2507** (Qwen et al., 2025), **DeepSeek-R1-Distilled-Llama-8B** (DeepSeek-AI et al., 2025) and **Phi-4-reasoning** (Abdin et al., 2025).

Table 1: Performance comparison of various reasoning methods on mathematical, scientific, general, and coding benchmarks. "Acc" denotes accuracy, "Tok" denotes token count, and "Lat" denotes total end-to-end latency. ↑ indicates that higher values are better, while ↓ indicates that lower values are better. For the early-exit methods (NoThink, DEER, and SpecExit*), the highest and second-highest Acc values are marked in **bold** and underline, respectively. Across all methods, the smallest and second-smallest Tok and Lat values are marked **bold** and underline, respectively. SpecExit* uses default parameter settings consistent with the best variants in ablation studies.

| | Math | | | | | | | | | Coding | | | Science | | | Logic | | |
| --- | --- | --- | --- | --- | --- | --- | --- | --- | --- | --- | --- | --- | --- | --- | --- | --- | --- | --- |
| | GSM8K | | | MATH500 | | | AIME | | | HUMANEVAL+ | | | GPQA-D | | | ARC-Challenge | | |
| Method | Acc↑ | Tok↓ | Lat↓ | Acc↑ | Tok↓ | Lat↓ | Acc↑ | Tok↓ | Lat↓ | Acc↑ | Tok↓ | Lat↓ | Acc↑ | Tok↓ | Lat↓ | Acc↑ | Tok↓ | Lat↓ |
| *Qwen3-4B-Thinking-2507* | | | | | | | | | | | | | | | | | | |
| *Vanilla* | 95.3 | 1414 | 155.6 | 96.6 | 6719 | 530.1 | 86.7 | 19577 | 243.3 | 90.9 | 5079 | 175.3 | 68.7 | 9041 | 325.8 | 95.6 | 1812 | 156.5 |
| *NoThink* | **95.2** | 1631 | 204.2 | 96.6 | 6395 | 488.5 | 86.7 | 19816 | 243.2 | 88.4 | 4480 | 131.5 | 67.2 | 8833 | 276.8 | **95.1** | 1889 | 159.8 |
| *DEER* | 94.3 | 960 | 230.3 | 94.4 | 4893 | 519.6 | 70 | **17838** | 218.6 | 86.6 | **4079** | 242.4 | 67.2 | 9053 | 505.2 | 94.6 | 1011 | 200.3 |
| *EAGLE3* | 94.8 | 1408 | 140.3 | 96.6 | 6670 | 395.7 | 80 | 19792 | 206.1 | 87.2 | 5178 | 81.7 | 67.7 | 8975 | 212.2 | 95.7 | 1822 | 164.2 |
| *SpecExit** | 93.8 | 649 | **75.8** | 96.8 | 4777 | 367.9 | 90 | 17769 | 187.3 | 89.6 | 4319 | 58.4 | 68.7 | 7011 | 137 | 94.5 | **588** | **71.4** |
| *DeepSeek-R1-Distill-Llama-8B* | | | | | | | | | | | | | | | | | | |
| *Vanilla* | 76.4 | 1008 | 629.4 | 81.8 | 6878 | 857.1 | 36.7 | 22170 | 307 | 74.4 | 6287 | 445.5 | 43.6 | 8857 | 574 | 49.9 | 1917 | 628.5 |
| *NoThink* | 54.6 | **233** | **22.2** | 55.2 | **1643** | **262.8** | 10 | 8744 | 184.1 | 46.3 | **472** | **7.3** | 26.8 | **1200** | **166.6** | 12.6 | **135** | **13.6** |
| *DEER* | 74.7 | 710 | 484.8 | **80.8** | 3533 | 973.3 | **40** | 15619 | 272.3 | 79.3 | 4206 | 269.2 | 40.9 | 8492 | 521.5 | 47.5 | 1029 | 531.3 |
| *EAGLE3* | 79.3 | 976 | 276.9 | 80.8 | 6172 | 593.6 | 30 | 25686 | 228.1 | 78.7 | 5312 | 346.5 | 43.9 | 8749 | 420.1 | 59.2 | 1378 | 496.4 |
| *SpecExit** | 75.3 | 333 | 112.6 | 80.6 | 1968 | 348.3 | 36.7 | **8160** | 176 | 81.7 | 3105 | 118.1 | 46 | 6849 | 307.5 | 50.3 | 500 | 253.7 |
| *Phi-4-reasoning* | | | | | | | | | | | | | | | | | | |
| Vanilla | 95.8 | 709 | 207.2 | 94.9 | 2122 | 543.7 | 74.2 | 10980 | 536.3 | 72.6 | 2059 | 300.2 | 68.7 | 7544 | 726.7 | 96.6 | 607 | 193.2 |
| NoThink | 95.7 | 668 | 197.3 | **94.1** | 2051 | 554.8 | 70.7 | 11104 | 509.4 | **72.8** | 1919 | 297.3 | 64.7 | 7334 | 710.0 | **96.7** | 588 | 178.0 |
| DEER | 95.5 | 582 | 223.5 | 92.4 | **1502** | 516.0 | 60.0 | **7003** | 507.4 | 66.3 | **1420** | 211.1 | 65.2 | **4479** | 1296.1 | 96.0 | 540 | 183.4 |
| EAGLE3 | 95.2 | 707 | 153.1 | 94.5 | 2136 | 324.2 | 74.0 | 10657 | 308.5 | 72.0 | 2035 | 155.8 | 68.2 | 7512 | 478.2 | 96.7 | 615 | 128.9 |
| *SpecExit** | **95.8** | 400 | **61.0** | 93.6 | 1750 | 271.9 | 74.7 | 9988 | 272.6 | 72.0 | 1605 | 131.8 | 67.7 | 6922 | 422.9 | 95.8 | 286 | 80.0 |

We compare our SpecExit method against several baselines: **Vanilla**, which represents full generation without any early-exit mechanism; **NoThink** (Ma et al., 2025a), which skips the reasoning phase; **DEER** (Yang et al., 2025), a dynamic early-exit method; and **EAGLE3** (Li et al., 2025), a speculative decoding baseline. For a fair and consistent comparison, the speculative decoding component in our system adopts the same draft-model architecture as EAGLE3, namely a one-layer causal model whose hidden size matches the corresponding target model. The draft model is trained together with SpecExit signals using the same training procedure to ensure comparable conditions.

Our performance analysis is based on three key metrics, as detailed in Table 1: **Accuracy** (↑), **Token** (↓) count and end-to-end **Latency** (↓). All experimental results are obtained by implementing our early-exit strategy in vLLM (Kwon et al., 2023), and running inference on an 8×H20 GPU cluster.

## 4.2 MAIN RESULTS

We first evaluate the proposed SpecExit against baseline reasoning approaches on mathematical, scientific, coding, and logical benchmarks. As shown in Table 1, SpecExit consistently achieves substantial reductions in both output length and inference latency while maintaining comparable or even higher accuracy.

Across benchmarks, SpecExit significantly shortens reasoning traces, with up to 54% and 53% reduction on GSM8K and ARC-Challenge for Qwen3-4B-Thinking-2507, and up to 66% and 64% reduction for DeepSeek-R1-Distill-Llama-8B. The reduced reasoning length corresponds to measurable efficiency improvements: SpecExit achieves a up to 1.9x latency reduction with Qwen3-4B-Thinking-2507 and up to 2.5x speedup with DeepSeek-R1-Distill-Llama-8B on GSM8K, compared with the speculative decoding baseline EAGLE3. Importantly, these gains come only with marginal

accuracy differences, confirming that early termination of redundant reasoning does not harm task performance. By contrast, prior inference-time methods primarily focus on reducing output length, but the latency gains they achieve are relatively modest. In some datasets, the additional computational overhead even leads to slower inference than the standard think mode. Notably, for Qwen3-4B-Thinking-2507 and Phi-4-reasoning models, inserting the `</think>` token at the beginning of reasoning in the NoThink baseline still fails to suppress reasoning, yielding output lengths similar to the vanilla Think mode and occasionally slightly longer.

Overall, these results demonstrate that SpecExit achieves a favorable balance between efficiency and accuracy, highlighting the practicality of integrating reasoning-aware early-exit strategies into LRMs inference.

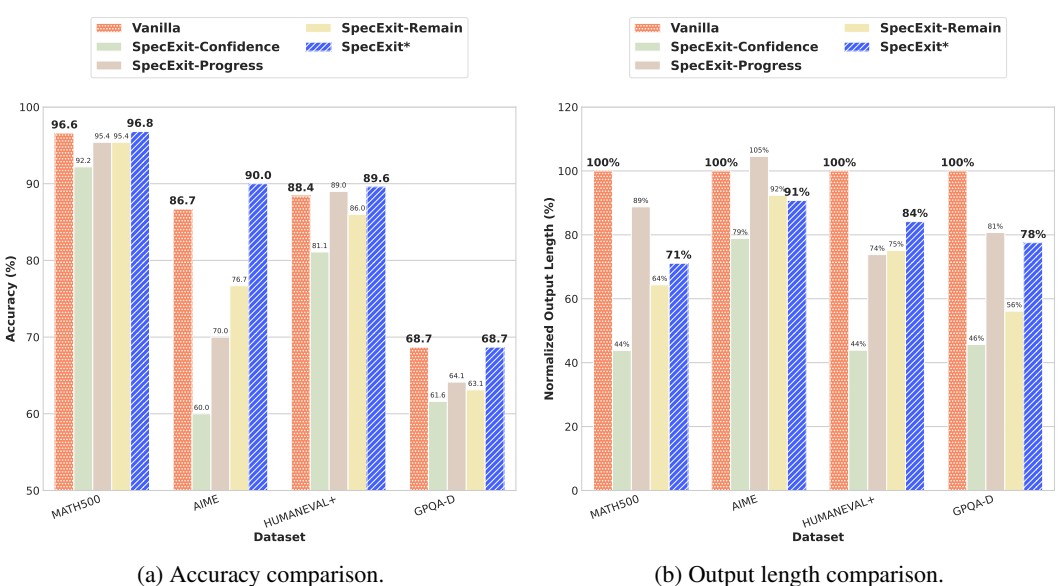

(a) Accuracy comparison.    (b) Output length comparison.

Figure 5: Ablation study of SpecExit signal types on Qwen3-4B-Thinking-2507.

### 4.3 ABLATION STUDY

**Signal Type**. To investigate the impact of individual reasoning signals in SpecExit, we conduct ablation studies on confidence, progress, and remaining token length, along with a combined configuration (SpecExit*) that integrates all three. As shown in Figure 5, the confidence-only variant yields the largest token reduction but overestimates the model's certainty, resulting in noticeable accuracy drops on complex benchmarks. The predicted reasoning progress increases sharply in the early steps yet continues to fluctuate during iterative reflection. Remaining token length is generally high at the beginning of inference but often triggers premature exits on complicated problems. By integrating all signals, SpecExit* leverages their complementary strengths, preserving competitive accuracy while substantially reducing tokens, demonstrating that multi-signal integration mitigates individual biases and enables more reliable early stopping.

**Signal Smoothing**. In order to investigate the influence of different smoothing strategies on the stability and performance of early-exit decisions, we conducted a series of ablation experiments comparing multiple approaches. As shown in Table 2, removing smoothing increases the volatility of cognitive signals, leading to inconsistent early exits and increased token consumption. The momentum-based prediction strategy significantly reduces token usage, though it may slightly degrade accuracy due to overly aggressive early termination. Smoothing using sliding-window and paragraph-level averaging offers a better trade-off, maintaining accuracy while improving efficiency. Among all methods, Exponential Weighted Moving Average (EWMA) strikes the most consistent balance, providing both stability and reliability. These results demonstrate that appropriate smoothing is essential for reliable early-exit behavior, as it mitigates the influence of transient fluctuations in the raw cognitive signals.

Table 2: Ablation study of different signal smoothing methods on Qwen3-4B-Thinking-2507. The highest Acc values are marked in bold for the early-exit methods.

| Method | MATH500 | | AIME | | HUMANEVAL+ | | GPQA-D | | Average | |
|---|---|---|---|---|---|---|---|---|---|---|
| | Acc↑ | Tok↓ | Acc↑ | Tok↓ | Acc↑ | Tok↓ | Acc↑ | Tok↓ | Acc↑ | Tok↓ |
| *Vanilla* | 96.60 | 6719 | 86.67 | 19577 | 88.40 | 5133 | 68.69 | 9041 | 85.09 | 10118 |
| *NoSmooth* | 94.20 | 3608 | 73.33 | 17832 | 92.10 | 2789 | 62.12 | 4066 | 80.44 | 7074 |
| *Momentum* | 91.80 | 2230 | 60.00 | 12427 | 83.50 | 2219 | 64.65 | 3406 | 74.99 | 5071 |
| *Sliding Window* | 95.20 | 4444 | 80.00 | 19184 | 86.60 | 4342 | 62.12 | 4738 | 80.98 | 8177 |
| *Paragraph Mean* | 95.40 | 4285 | 76.67 | 18231 | 84.80 | 4569 | 65.66 | 4726 | 80.63 | 7953 |
| *SpecExit\* (EWMA)* | **96.80** | 4777 | **90.00** | 17769 | **89.60** | 4319 | **68.69** | 7011 | **86.27** | 8469 |

**Step Split Tokens**. To evaluate the influence of different step split strategies on early-exit performance, we conducted ablation experiments comparing paragraph delimiters, general discourse markers, and a contrastive subset of discourse markers. Discourse markers indicate semantic transitions or reasoning shifts, but are dependent on the underlying data and model, limiting their generality. In prior work on dynamic early-exit methods, contrastive subsets of discourse markers (e.g., "Wait", "But", "Alternatively") are frequently used to capture reasoning-relevant transitions. In contrast, paragraph delimiters (\n\n) provide a more general segmentation that does not rely on model-specific or dataset-specific patterns. As shown in Figure 6, using paragraph delimiters achieves competitive accuracy and token reduction, demonstrating that a general segmentation strategy can be effective for early-exit decisions while maintaining coherence in reasoning traces.

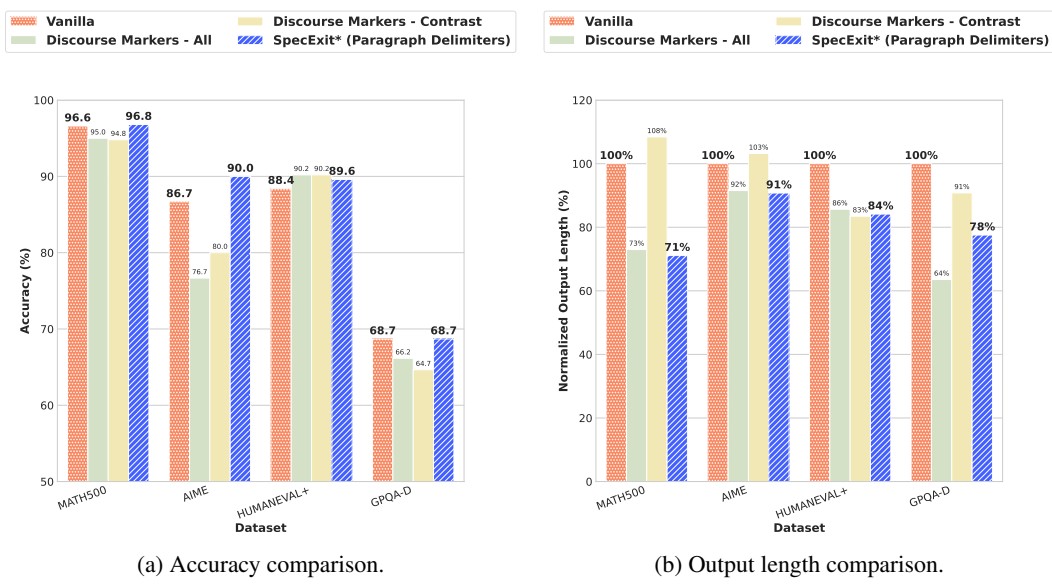

(a) Accuracy comparison.  (b) Output length comparison.

Figure 6: Ablation study of step split tokens strategies on Qwen3-4B-Thinking-2507.

In summary, our ablation studies on signal types, smoothing strategies, and step split methods provide key insights for improving early-exit decision-making. The integration of multiple signals strikes the best balance between accuracy and token efficiency, while appropriate smoothing methods stabilize cognitive signals and enhance the consistency of early exits. Additionally, using general segmentation strategies, such as paragraph delimiters, improves the generalizability of early-exit systems across diverse datasets. These findings emphasize the importance of a holistic approach, where complementary strategies jointly enhance both efficiency and reliability.

## 5 RELATED WORK

**Efficient Reasoning.** To mitigate unnecessary CoT generation in LRMs (Chen et al., 2025; Sui et al., 2025), prior work has explored both training-based and inference-time strategies. Training-based approaches typically modify model behavior through reinforcement learning with length-sensitive objectives (Aggarwal & Welleck, 2025; Yeo et al., 2025; Shen et al., 2025) or supervised fine-tuning on reasoning traces of varying lengths (Ma et al., 2025b; Munkhbat et al., 2025). While effective in shortening outputs, these methods demand substantial retraining cost and can distort the model's output distribution, raising concerns about reliability and generalization to unseen tasks. Inference-time methods avoid retraining and instead attempt to stop reasoning dynamically by monitoring model signals such as logits (Yang et al., 2025) or intermediate answers (Fu et al., 2024). Although these methods show that early stopping can reduce reasoning length without degrading accuracy, their reliance on probing introduces additional computation and often emphasizes token count reduction rather than true end-to-end latency improvements.

**Speculative Decoding and Hidden States.** Speculative decoding (Chen et al., 2023; Leviathan et al., 2023) is a widely adopted technique for accelerating decoding speed, where a lightweight draft model proposes candidate tokens that a larger target model verifies in a single pass. Recent methods (Cai et al., 2024; Li et al., 2024a;b; 2025; Zhang et al., b) leverage hidden states to predict multiple future tokens. Beyond speculative decoding, several studies (Lin et al.; Zhang et al., c; Dong et al.; Zhang et al., a), have revealed that hidden states contain broader information about future outputs, including correctnes, response length, and reasoning paths. Building on this insight, our method extends speculative decoding by training hidden states not only to forecast future tokens but also to produce an early-exit signal.

## 6 CONCLUSION

In this work, we propose **SpecExit**, a reasoning-aware early-exit framework that leverages latent signals from models' hidden states to dynamically terminate reasoning processes in LRMs. By concatenating auxiliary prediction heads to a lightweight draft model, SpecExit simultaneously predicts future tokens and early-exit signals in a single forward pass, eliminating the probing overhead required by previous approaches. Our experiments across diverse tasks and models demonstrate that SpecExit substantially reduces reasoning length by up to 66% and achieves significant end-to-end latency improvements up to 2.5x without compromising accuracy. The proposed method highlights the potential of hidden states as informative signals for efficient reasoning and establishes a practical pathway for deploying LRMs in real-world scenarios.

## 7 ETHICS STATEMENT

This research does not involve human subjects, sensitive personal data, or applications with foreseeable negative societal impact. All datasets mentioned are publicly available, and proper licenses and usage guidelines are respected.

## 8 REPRODUCIBILITY STATEMENT

We have taken extensive measures to ensure the reproducibility of our results. The code used to implement our proposed framework, SpecExit, is publicly available at: `https://anonymous.4open.science/r/SpecExit-B802` . Detailed instructions on how to use and run the code, including environment setup and dependency installation, are provided in the repository. For the experiments conducted in this paper, we used publicly available benchmark datasets and models.

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

# A APPENDIX

This section may contain supplementary materials such as additional experimental details, ablation studies, hyperparameter settings, and qualitative examples of generated CoT sequences.

## A.1 USAGE OF LLMS

We used LLMs(ChatGPT, Gemini) for grammar reviews and style polishing.

## A.2 SUPPLEMENTARY EXPERIMENT

**Analysis of Hidden States:** To investigate whether intermediate hidden representations encode discriminative signals relevant to reasoning sufficiency, we analyze hidden states extracted from multiple depth stages of the reasoning traces. We apply Principal Component Analysis (PCA) and Uniform Manifold Approximation and Projection (UMAP) to project the hidden states into lower-dimensional spaces and analyze their spatial distribution. The analysis reveals that, across datasets of varying difficulty, the hidden states at the "start," "mid," and "end" positions exhibit clear clustering patterns. Moreover, the hidden states at the start of the reasoning trace show notable cross-dataset similarity, as illustrated in Figure 7a. In addition, when examining the relationship between the embedding representations of hidden states and the paragraph geometric mean of probabilities, we observe that hidden states also form meaningful clusters under different probability ranges, as illustrated in Figure 7b.These observations suggest that hidden representations indeed capture information related to dataset difficulty, reasoning progress, paragraph-level average probability, thereby providing preliminary evidence that functions of intermediate hidden states can serve as reliable proxies for reasoning sufficiency.

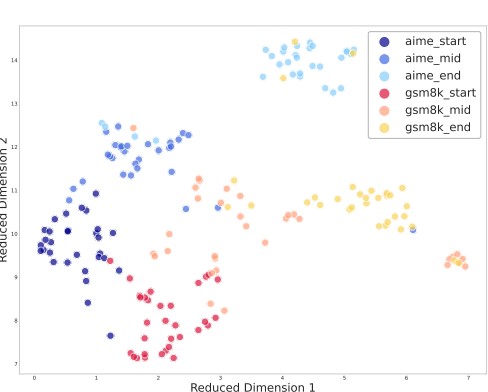 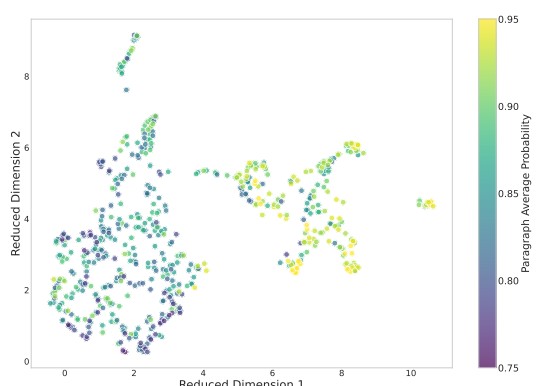

(a) Distribution of hidden states on the GSM8K and AIME datasets. The labels "start," "mid," and "end" correspond to early, intermediate, and final positions along the reasoning trace, respectively.

(b) Relationship between the embedding representations of hidden states and the paragraph geometric mean of probabilities, showing clustering patterns under different confidence ranges.

Figure 7: Analysis of hidden-state representations on the GSM8K and AIME datasets. (a) shows the distribution of hidden states along different reasoning positions; (b) shows the relationship between hidden states and paragraph-level confidence signals.

**Signal Type:** In Figure 8, we conduct a systematic comparison of different early-exit signal configurations, including the full combination of three signals (confidence, reasoning progress, and remaining reasoning length) and the reduced configuration using only "progress + remaining." The results show that relying solely on progress and remaining signals can reduce the reasoning length to some extent, but this setting consistently underperforms the full three-signals configuration in accuracy, especially on datasets involving longer reasoning chains or higher task complexity. In contrast, the complete three-signals design exhibits more stable behavior across datasets, effectively shortening the output length while maintaining a more favorable accuracy-efficiency trade-off. Overall, these findings demonstrate that incorporating the confidence signal is essential for constructing stable and

generalizable early-exit strategies, enabling more consistent efficiency gains and reliable correctness compared with the "progress + remaining" setting.

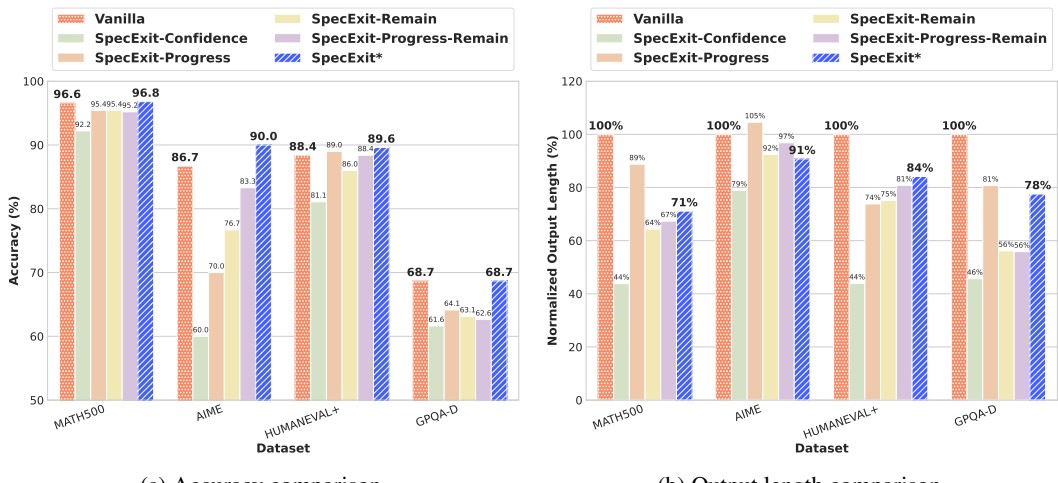

(a) Accuracy comparison.  (b) Output length comparison.

Figure 8: Ablation study of SpecExit signal types on Qwen3-4B-Thinking-2507, adding experiment with only Progress and Remaining signals.

**Threshold Calibration:**

In the ablation study of SpecExit signal types, the following thresholds are applied as stopping conditions for the respective signal types: SpecExit-Confidence requires predicted confidence value greater than 0.9, SpecExit-Progress requires predicted progress value greater than 0.8, SpecExit-Remain requires predicted remaining reasoning length value less than 100, and SpecExit* combines thresholds with a predicted confidence value greater than 0.8, predicted progress greater than 0.3, and predicted remaining reasoning length less than 200.

In addition to the thresholds reported above, we provide two complementary procedures that further calibrate the stopping criteria using a small held-out calibration set. Specifically, we sample 90 instances of varying difficulty from the validation split of training data and conduct the following analyses:

(1) Statistical distribution-based thresholding. During data construction, we have access to the shortest valid reasoning path for each problem, and thus we can determine whether stopping at the end of any intermediate paragraph would still yield a correct final answer. By examining the empirical distribution of the predicted signals at these paragraph boundaries and correlating them with correctness, we obtain the distributions shown in Figure 10. These distributions allow us to derive a signal-specific threshold that maximizes the retention of correct answers under early stopping.

(2) Design space exploration over the threshold search space. We additionally perform a design space exploration (DSE) over a predefined grid of confidence, progress, and remaining-length thresholds. Using vLLM to run this evaluation pipeline on the 90-sample calibration set, the full search requires approximately 2.5 hours on an 8×H20 GPU cluster. Among all threshold combinations, we select those lying on the Pareto frontier that best trade off accuracy preservation against reduction in reasoning length. The results are shown in Figure 10.

### A.3 IMPLEMENTATION DETAILS

**Model Architectures:** We evaluate SpecExit on three large reasoning models (LRMs): **Qwen3-4B-Thinking-2507** (Qwen et al., 2025), **DeepSeek-R1-Distilled-Llama-8B** (DeepSeek-AI et al., 2025) and **Phi-4-reasoning** (Abdin et al., 2025). For speculative decoding, the draft models adopt the EAGLE3 (Li et al., 2025) architecture. The draft models are single-layer causal models whose hidden sizes match those of the corresponding target models. The input embedding layer of each draft model is shared with its corresponding target model to ensure tokenizer compatibility, while the

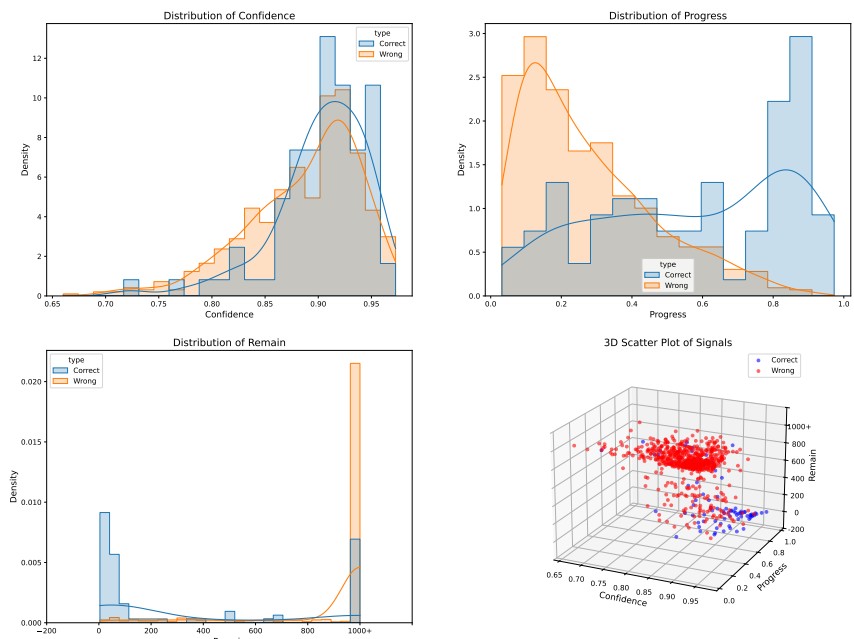

Figure 9: Distribution of signals with correct and wrong answers in calibration data.

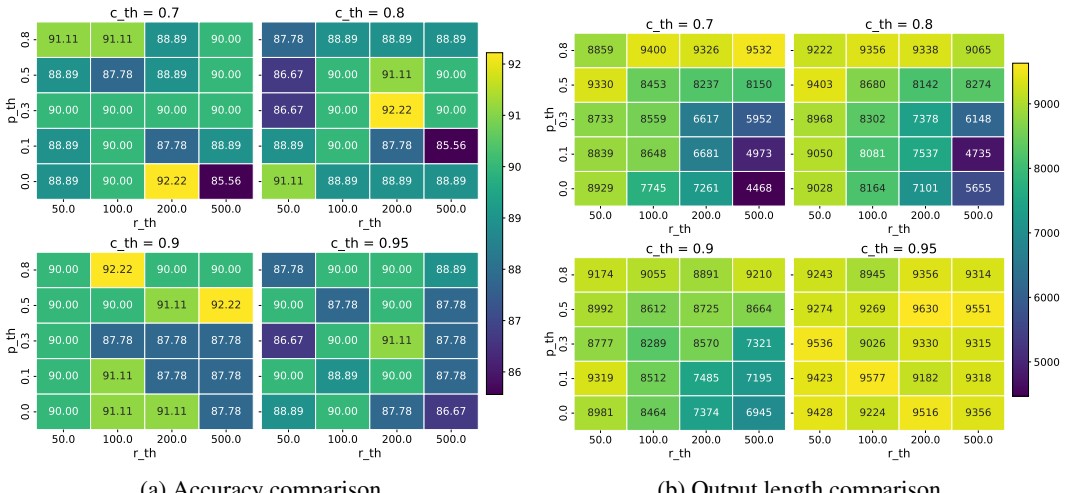

(a) Accuracy comparison.  (b) Output length comparison.

Figure 10: Accuracy and output length comparison in the calibration data under different threshold settings. The parameters c_th, p_th, and r_th denote the thresholds for confidence, progress, and remaining reasoning length estimation, respectively.

output head uses a compact vocabulary of 32k high-frequency tokens. The detailed configurations are summarized in Table 3.

Figure 11 provides an overview of how SpecExit integrates with the EAGLE3 speculative decoding pipeline. During inference, the draft model produces multi-level token predictions, and simulta-neously, our early-exit module tracks the evolution of three reasoning-related signals to determine whether the target model can safely terminate the thinking phase. The design is architecture-agnostic and can be combined with other multi-token-prediction frameworks such as Medusa. In practical deployment, the fully-connected layer of the MTP head (e.g., EAGLE3 or Medusa) can be fused with the early-exit module, effectively hiding additional operator invocation latency.

Table 3: Target and Draft Model Configurations.

| Model Name | Role | Architecture | Hidden Size | Layers | Vocab Size |
|---|---|---|---|---|---|
| Qwen3-4B-Thinking-2507 | Target | Qwen3ForCausalLM | 2560 | 36 | 151936 |
| | Draft | Eagle3LlamaForCausalLM | 2560 | 1 | 32000 |
| DeepSeek-R1-Distilled-Llama-8B | Target | LlamaForCausalLM | 4096 | 32 | 128256 |
| | Draft | Eagle3LlamaForCausalLM | 4096 | 1 | 32000 |

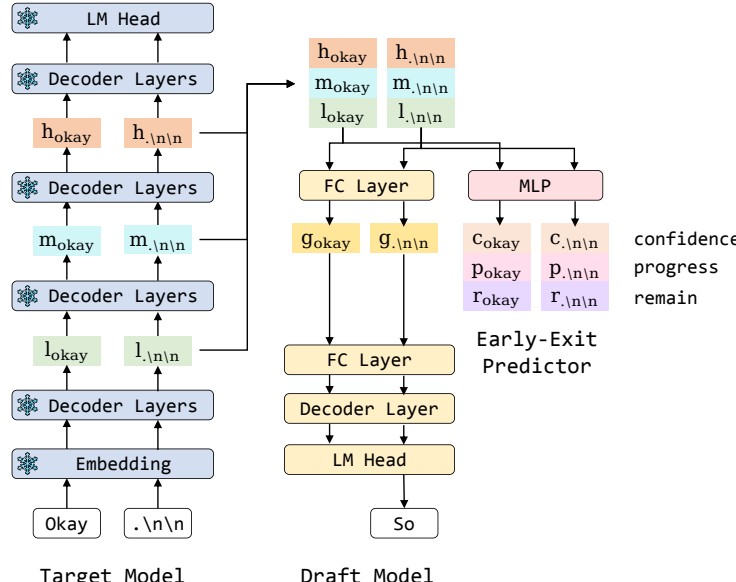

Figure 11: Diagram of the speculative decoding pipeline integrating EAGLE3 (Li et al., 2025) with the proposed reasoning early-exit mechanism.

**Signal Smoothing**: In the ablation study of signal smoothing strategies, the following methods are implemented to stabilize cognitive signals for early-exit decisions:

- Sliding Window: The sliding window approach smooths the signal by averaging the last $N$ predicted signal values, with $N$ set to 10. The mean score $x_t$ at decoding step $t$ is computed as:

$$x_t = \text{Mean}(s_t, N) = \frac{1}{N} \sum_{i=t-N+1}^{t} s_i, \tag{7}$$

where $s_i$ denotes the predicted signal value at decoding step $i$.

- Momentum-based Prediction: This method predicts the next score based on the momentum, which is calculated as the difference between $N$ consecutive signal values, with $N$ set to 10. The predicted score $x_t$ at decoding step $t$ is given by:

$$x_t = \text{Predict}(s_t, N) = s_{t-1} + \frac{1}{N-1} \sum_{i=t-N+1}^{t-1} (s_i - s_{i-1}). \tag{8}$$

- Paragraph Mean: In this approach, the score $x_t$ is calculated as the average of all predicted signal values within the current paragraph:

$$x_t = \frac{1}{T} \sum_{i=1}^{T} s_i, \tag{9}$$

where $T$ is the total number of steps in the current paragraph.

- Exponential Weighted Moving Average (EWMA): In this approach, the smoothing factor $\alpha$ is set to 0.1. The new score $x_t$ is updated based on the previous score $x_{t-1}$ and the current signal value $s_t$ as:

$$x_t = \text{EWMA}(s_t, x_{t-1}, \alpha) = \alpha \cdot s_t + (1 - \alpha) \cdot x_{t-1}. \tag{10}$$

**Discourse Markers**: We collect high-frequency words appearing at the beginning of model-generated sentences as discourse markers. As shown in Figure 12, we present examples of discourse markers extracted from Qwen3-4B-Thinking-2507 on the MATH dataset. Among them, transitional words such as "Wait" and "But" can be regarded as a subset of these high-frequency markers.

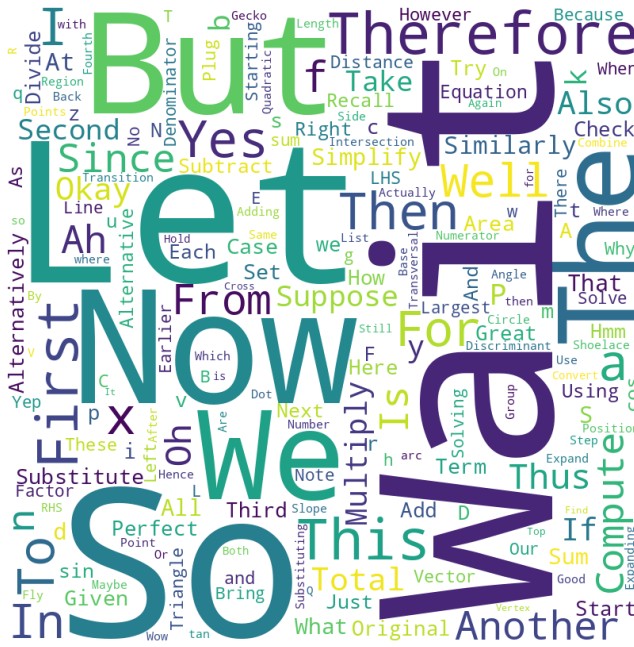

Figure 12: Discourse marker distribution in Qwen3-4B-Thinking-2507's responses on the MATH500 (Hendrycks et al., 2021) dataset.

### A.4 PREDICTED SIGNAL VISUALIZATION

As shown in Figure 13, for simple problems, the predicted confidence remains high, the predicted remaining reasoning length is relatively short, and the predicted progress rises rapidly within the first few sentences, with only minor drops on a few uncertain words. This indicates that for such problems, the model is able to establish a stable reasoning trace at an early stage. In contrast, as shown in Figure 14, for complicated problems, the model also exhibits high confidence and a short remaining reasoning length in the initial summarization phase, but once it enters the detailed analysis stage, the predicted remaining reasoning length increases significantly, confidence drops. Meanwhile, the predicted progress starts low and rises initially, but then fluctuates markedly during repeated self-reflection, making it difficult to stabilize at a high threshold.

These observations reveal the inherent limitations of relying on individual signals for early exiting. When depending solely on confidence, the model often exhibits overconfidence and terminates too early before sufficient reasoning has been completed, leading to substantial accuracy degradation. When depending solely on the predicted remaining reasoning length, the model may become overly optimistic in the early stages of complicated problems, resulting in premature exits before essential reasoning steps are accomplished. When depending solely on progress, the signal tends to fluctuate and remain unstable in complex reasoning tasks, making it difficult to trigger an appropriate early exit and thereby restricting achievable speedup. In summary, each single signal suffers from the inability to balance accuracy and efficiency across diverse problem types. By integrating multiple signals in a complementary manner, the model can achieve a smoother trade-off between reasoning accuracy and inference acceleration.

Figure 15 illustrates the effect of applying an Exponentially Weighted Moving Average (EWMA) to stabilize predicted signal. EWMA effectively suppresses local noise while preserving the global trend, leading to smoother traces for signal. The smoothed signals reveal clearer overall convergence patterns, providing a more stable and reliable basis for threshold-based early-exit decisions and thereby improving robustness in challenging reasoning scenarios.

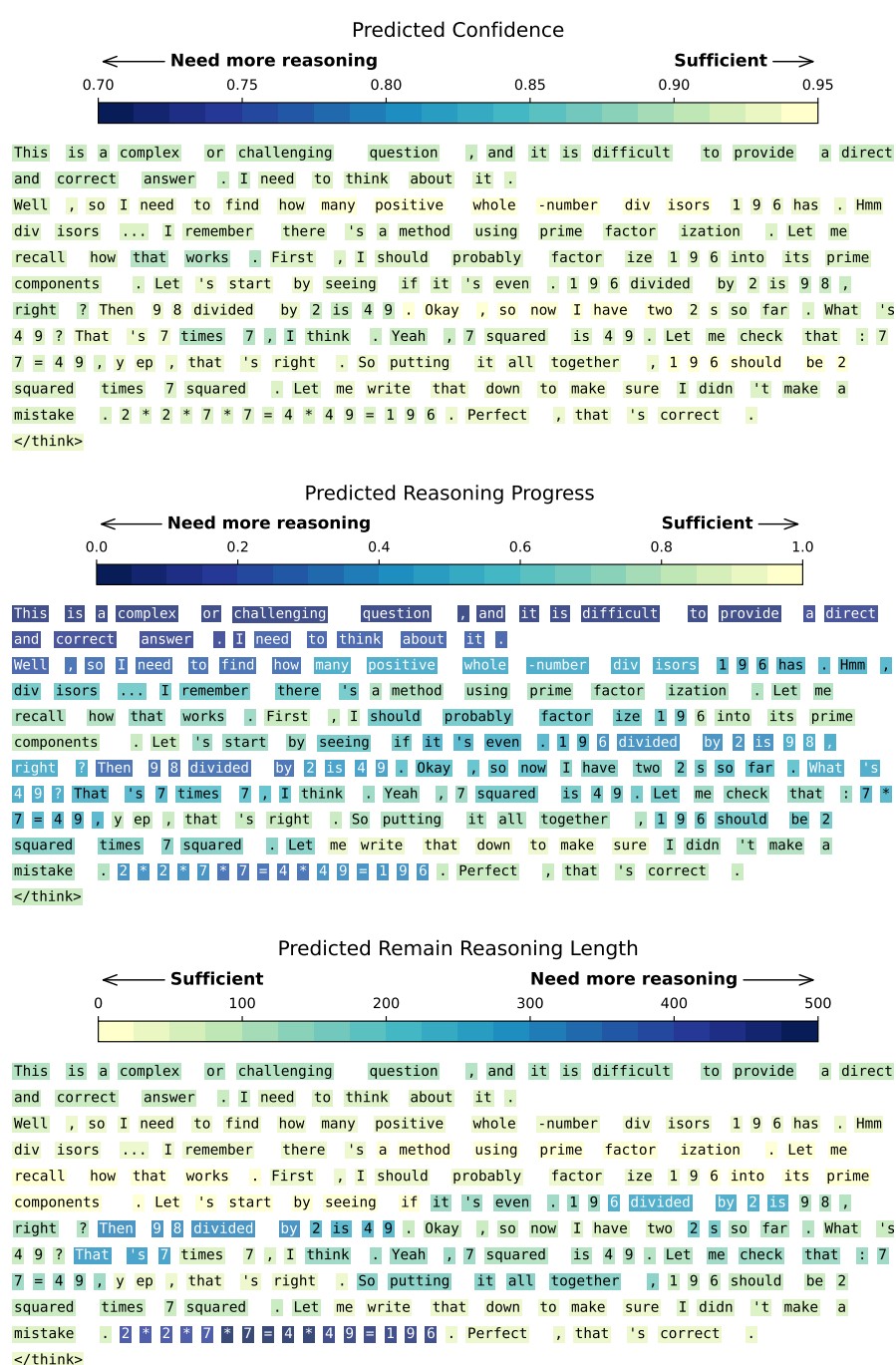

Figure 13: Visualization of reasoning signals for a simple problem, illustrated with an example from the MATH500 (Hendrycks et al., 2021) dataset, where darker colors denote insufficient reasoning and lighter colors denote sufficiency.

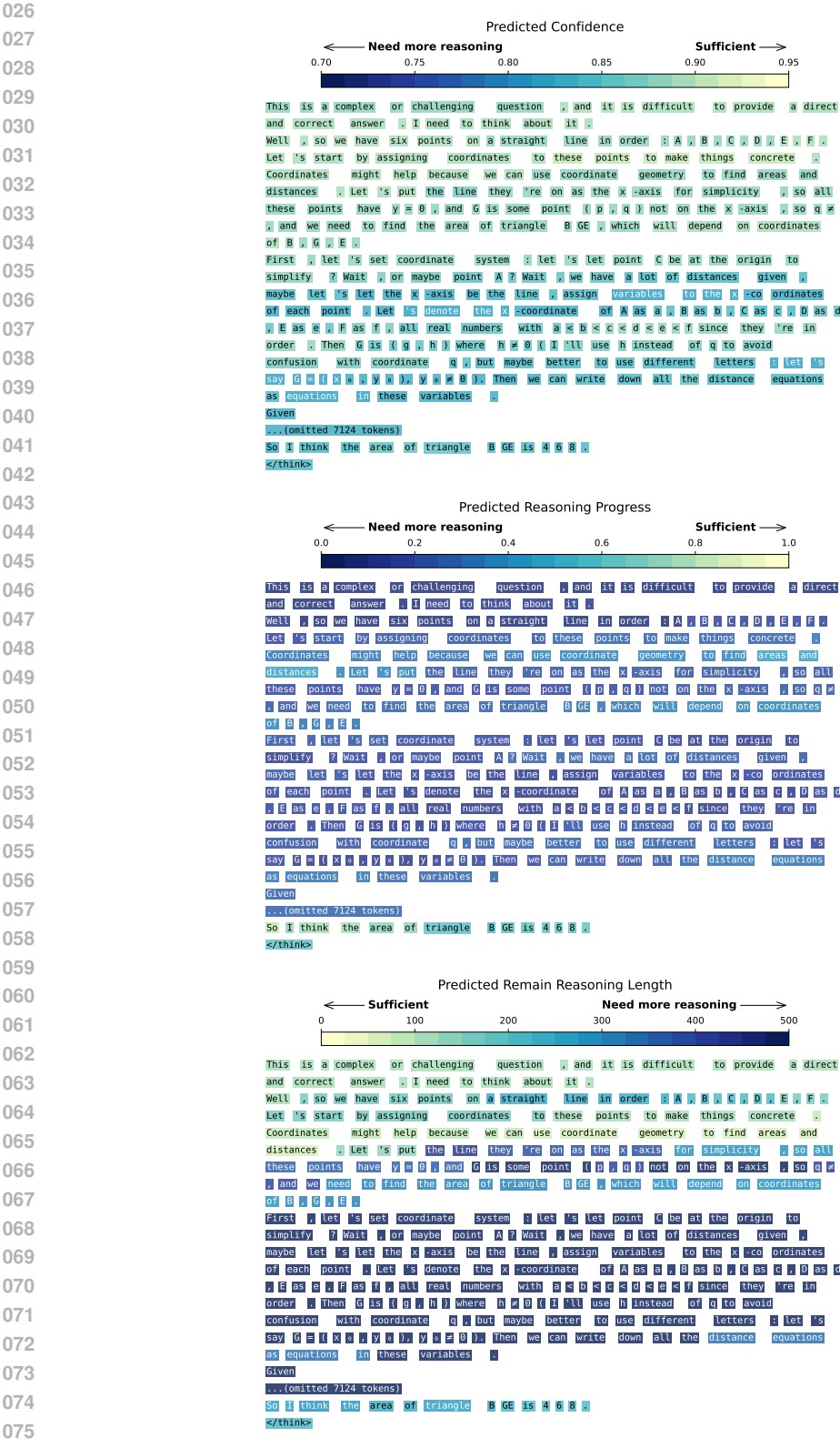

Figure 14: Visualization of reasoning signals for a complicated problem, illustrated with an example from the AIME (MAA Committees) dataset, where darker colors denote insufficient reasoning and lighter colors denote sufficiency.

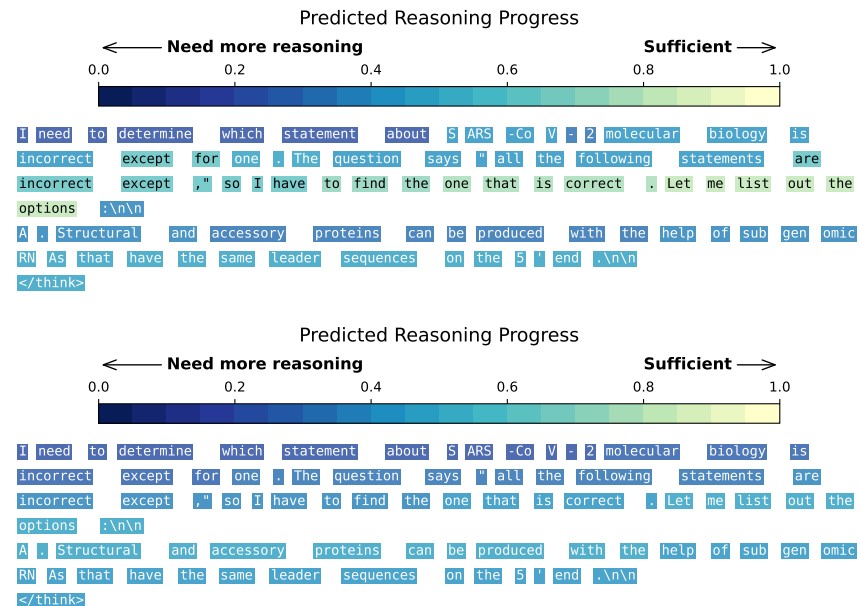

Figure 15: Visualization of the smoothing effect introduced by the EWMA-based signal estimator, illustrated with an example from the GPQA Diamond (Rein et al., 2023) dataset, where the stabilized trace reveals reduced high-frequency fluctuations and clearer monotonic trends in the underlying reasoning signals.

## A.5 CASE STUDY DETAILS

Figure 16 presents an example from the GSM8K (Cobbe et al., 2021) dataset, where SpecExit is applied after an initial analysis. In this case, SpecExit inserts a decision to exit reasoning based on the signal magnitude after completing the first paragraph, thus preventing the continuation of redundant reasoning tokens. The process begins with the model evaluating the initial segment of the problem, analyzing the available context and producing intermediate reasoning steps. When SpecExit detects that all the signals exceeds a predefined threshold, it triggers the early exit mechanism. As a result, the model halts further reasoning and transitions directly to the answering phase, thus producing the correct result in a more efficient manner.

By shortening the unnecessary token generation and reducing redundant reasoning, SpecExit leads to significant performance improvements in terms of inference speed while maintaining the correctness of the answer. This early exit strategy not only accelerates the inference process but also avoids the potential confusion that the model might experience from excessive reflection or unnecessary computations. This case highlights the effectiveness of integrating early exit mechanisms in enhancing both the efficiency and reliability of large language model inference.

**Question:**

Ram uses a lot of pens. He discovered that he can save money by mixing the ink from five empty pens to make one full pen. If he buys 25 pens and then uses them to make new pens when the ink runs low, how many total pens does he get to have?

**SpecExit:** **(421 tokens)**

Okay, let's try to figure out how many sterling silver beads Katerina uses per necklace. Hmm, first, let's list out what we know.\n\nShe uses 8 topaz gemstones per necklace. Each topaz is 1 inch long, so the total length of the topaz gemstones would be 8 times 1 inch, right? So that's 8 inches.\n\n</think>     **Signals: confidence=0.94 progress=41% remain_tokens=56** ✅

To determine how many sterling silver beads Katerina uses per necklace... (omitted)
### Final Answer\n\n$$\n\\boxed{68}\n$$\n\n
Katerina uses **68** sterling silver beads per necklace.     **correct** ⊘

**Vanilla: (1521 tokens)**     **Initial**

Okay, let's try to figure out how many sterling silver beads Katerina uses per necklace. Hmm, first, let's list out what we know.\n\nShe uses 8 topaz gemstones per necklace. Each topaz is 1 inch long, so the total length of the topaz gemstones would be 8 times 1 inch, right? So that's 8 inches.

Hmm, let's start by breaking down the problem step by step. \n\n     **Analyzation**
First, the problem says she uses 8 topaz gemstones per necklace, and each topaz is one inch long. So, the total length of the topaz gemstones would be 8 times 1 inch, right? Let me write that down: 8 topazes * 1 inch each = 8 inches.\n\n
Now, the necklace is supposed to be a total length of 25 inches. That means the space between the gemstones, which is filled with sterling silver beads, has to make up the rest of the length. So, the total length from the silver beads would be 25 inches minus the length of the topaz gemstones. Let me calculate that: 25 inches - 8 inches = 17 inches. So, the silver beads together are 17 inches long.

But each sterling silver bead is one-quarter of an inch long... (omitted)     **Reflection**
Wait, hold on. When you have beads between gemstones, do you need to consider how many gaps there are? Because ... (omitted)
Oh! That's a crucial point I might have missed. The problem says \"fills the space between gemstones using sterling silver beads.\" So, the silver beads are in the spaces between the topaz stones. So, if there are 8 topaz stones, there are 7 spaces between them.\n\n
Let me re-examine the problem statement...(omitted)
Wait, but the problem says \"each necklace is made to a total length of 25 inches\". So, total length = length of topaz + length of silver beads.\n\n
But ...(omitted) (continuous reflection over 26 paragraphs)

So, in the original problem, 8 topaz stones (8 inches), total necklace 25 inches, so 17 inches of silver beads. 17 / 0.25 = 68.\n\n     **Summary**
I think that's the answer. The key is that the problem says \"fills the space between gemstones using sterling silver beads\", so the total length of the beads is the total necklace length minus the topaz length, and then divide by the bead length.\n\n
I think the confusion might be about the number of gaps, but since the problem doesn't specify that each gap has a certain number of beads, we just take the total length of the beads and divide by the length per bead.\n\n
So, the answer should be 68.
</think>

Figure 16: Discourse marker distribution in Qwen3-4B-Thinking-2507's responses on the MATH500 (Hendrycks et al., 2021) dataset.

