# OpenReview forum: "SpecExit: Accelerating Large Reasoning Model via Speculative Exit"
_ICLR.cc/2026/Conference — Submitted to ICLR 2026_

### Official Review · Reviewer_8nc3 · 2025-10-17

**Soundness:** 3
**Presentation:** 3
**Contribution:** 3
**Rating:** 6
**Confidence:** 3

**Summary:**

This paper proposes SpecExit, an early-exit framework that integrates with speculative decoding to predict both future tokens and reasoning sufficiency signals from a lightweight draft model, achieving up to 2.5× faster inference and 66% shorter reasoning without accuracy loss.

**Strengths:**

- Elegant integration of early-exit prediction into speculative decoding with zero probing overhead.
- Uses hidden-state signals (confidence/progress/remaining) to dynamically determine reasoning sufficiency.
- Demonstrates consistent 2×–2.5× speedup and 66% token reduction without hurting accuracy.
- Framework is general, modular, and easily deployable on vLLM or PyTorch inference pipelines.
- Well-designed ablation studies confirm the effectiveness of multi-signal fusion and signal smoothing.

**Weaknesses:**

- Model-specific training requirement: The MTP-based signal extraction head must be retrained for each target model due to differences in hidden-state representations, limiting plug-and-play generality.
- Limited model family evaluation: Experiments are conducted only on two reasoning model families, leaving generalization to other architectures or general-purpose LLMs unclear.
- Signal & smoothing dependency: The method relies on multiple signals and smoothing strategies (e.g., EWMA) to remain stable, indicating insufficient robustness of raw signals.

**Questions:**

- Since the MTP and signal heads must be retrained for each model due to architectural differences, how costly is this process in practice? Could a lightweight or partially shared training strategy (e.g., freezing most of the MTP layer or distilling from a pretrained head) reduce the retraining overhead across models?
-  Can the authors provide results or at least discussion on how SpecExit might generalize to other model architectures, such as Phi, Mistral, or LRMs?
- How sensitive is SpecExit’s performance to the chosen thresholds for confidence, progress, and remaining length? like use different thresholds to do some ablation studies. Why do you choose these thresholds in STEER method (like confidence > 0.8 && progress > 0.3 && remaining < 200)?
- Can the authors identify and analyze typical scenarios where the control signals conflict (e.g., rising confidence but low progress), and show how the EWMA controller behaves in these situations?

---

> ### Author Response · Authors · 2025-11-19
> **Response to Reviewer 8nc3 (Part 1/2)**
>
> We thank the reviewer for the insightful and valuable comments. We respond to each comment as follows and sincerely hope that our responses could properly address your concerns. We have updated our manuscript and highlighted the revisions for ease of inspection.
>
> > Question 1: Since the MTP and signal heads must be retrained for each model due to architectural differences, how costly is this process in practice? Could a lightweight or partially shared training strategy (e.g., freezing most of the MTP layer or distilling from a pretrained head) reduce the retraining overhead across models?
>
> We appreciate your insightful question regarding the retraining cost of the MTP and signal heads. Because the regression objectives of the exit-signal heads are orthogonal to the token-prediction objective of speculative decoding, these heads can either be trained jointly with the speculative decoding model or be initialized from an existing checkpoint that has already been trained for speculative decoding rather than training from scratch. Empirically, training one epoch takes approximately **1.5 hours (12 H20 GPU hours)**, and the early-exit signals typically converge within **1–3 epochs**.
>
>
>
> > Question 2: Can the authors provide results or at least discussion on how SpecExit might generalize to other model architectures, such as Phi, Mistral, or LRMs?
>
> Thank you for this insightful question. SpecExit can be extended to new model families as long as two conditions are satisfied:
>
> - (1) The model uses a dedicated special token as the “stop thinking” marker (e.g., `</think>`)
> - (2) There exists a token that can reasonably serve as a paragraph separator (e.g., `\n\n`, `.\n\n`)
>
> Adding support for a new model architecture follows the steps below:
>
> - Model definition and configuration: Add the model definition file that specifies how to access hidden states and how to construct the tree mask; Add the configuration file that records hyperparameters such as hidden size, number of heads, etc.
>
> - Data Construction: On the reasoning training set, we first obtain the responses from the new model to train draft model. For each response, we process the thinking portion by inserting the </think> token at paragraph boundaries and then checking whether the resulting answer is still correct, in order to extract the shortest valid reasoning trace.
>
> - Training SpecExit: We then train the draft model and early-exit signals. The training code itself does not require special modifications; one only needs to specify the stop-thinking token so that the loss for signals can be computed correctly.
>
> - Inference and threshold selection: During inference, we configure the stop-thinking token and the paragraph-separator token, and determine the confidence threshold following the procedure described in our answer to Question 3.
>
> Part of this process is the same with how EAGLE3 adds support for new model architectures.
>
> We have already released our anonymous code repository which is linked in the abstract, and we are actively working on improving the usability of the framework and adding support for more models, which will be gradually open-sourced.

---

> > ### Comment · Reviewer_8nc3 · 2025-11-26
> >
> > Question1: Thank you for the reply. While it is helpful to know that the MTP head is inexpensive to train, the fact remains that it cannot be reused across architectures or model sizes and must be retrained for each model. Moreover, it is still unclear whether the introduction of speculative decoding has any impact on downstream performance or inference stability.
> >
> > Question2: Thank you also for the detailed implementation steps. However, the response primarily describes how one would technically add support for another model family, rather than providing empirical evidence or discussion showing that SpecExit actually works on models such as Phi, Mistral, or LRMs. Even a qualitative comparison or small-scale evaluation would help substantiate the claimed generalizability of the approach.
> >
> > At this point, I would like maintain my socre.

---

> ### Author Response · Authors · 2025-11-19
> **Response to Reviewer 8nc3 (Part 2/2)**
>
> > Question 3: How sensitive is SpecExit’s performance to the chosen thresholds for confidence, progress, and remaining length? like use different thresholds to do some ablation studies. Why do you choose these thresholds in STEER method (like confidence > 0.8 && progress > 0.3 && remaining < 200)?
>
> We sincerely appreciate your insightful question regarding the sensitivity of SpecExit to the specific threshold choices used in the default configuration, as well as your suggestion of conducting ablation studies under alternative threshold settings.
>
> To address this, we have substantially expanded our analysis in Appendix A.2, where we introduce two approaches for determining the thresholds: one grounded in the statistical separation of signal values between correct and incorrect reasoning traces, and the other based on a systematic search for optimal threshold combinations. Specifically, we sample 90 instances of varying difficulty from the validation split of training data and design two methods of calibration procedures based on this held-out set.
>
> First, we examine the statistical behavior of confidence, progress, and remaining-length signals under “safe stopping” conditions. During dataset construction, we have access to each example’s shortest valid reasoning path, which allows us to determine whether stopping at any intermediate step yields the correct final answer. By correlating these safe-stopping events with the corresponding values of the three signals, we obtain the empirical distributions illustrated in **Figure 9 in Appendix A.2**.
>
> Second, to quantify robustness beyond the default choices, we perform a design space exploration (DSE) over the full three-dimensional threshold configuration space. Using a vLLM-based evaluation pipeline, we run an exhaustive grid search over a held-out calibration set of 90 instances, covering diverse reasoning difficulties. This grid search evaluates $4*5*4$ types of threshold combinations and identifies those lying on the Pareto frontier that optimally balance accuracy preservation and reduction in reasoning length, as shown in **Figure 10 in Appendix A.2**.
>
> These additional results and analyses have been incorporated into the **Appendix A.2** to directly address your insightful comments.
>
>
>
>
> > Question 4: Can the authors identify and analyze typical scenarios where the control signals conflict (e.g., rising confidence but low progress), and show how the EWMA controller behaves in these situations?
>
>
> We appreciate your valuable suggestion.
>
> For the first question, **Figure 14 in Appendix A.3** illustrates a representative and interesting case. For complex reasoning tasks, the model exhibits high confidence and a short predicted remaining reasoning length during the initial summarization phase, but once it enters the detailed analytical stage, the predicted remaining reasoning length rises substantially while confidence decreases. Meanwhile, the predicted progress starts low and increases initially, but later fluctuates significantly during repeated self-reflection, making it difficult for the signal to stabilize at a high threshold. This example highlights a typical pattern of conflict among control signals along challenging reasoning trajectories.
>
> For the second question, **Figure 15 in Appendix A.3** illustrates the effect of EWMA. EWMA effectively suppresses local noise while preserving the global trend, leading to noticeably smoother trajectories for confidence, reasoning progress, and remaining reasoning length. The smoothed signals reveal clearer convergence patterns and offer a more stable basis for exit decisions when raw signals exhibit conflicting behavior.
>
> Importantly, we would like to clarify that the use of these signals is a deliberate design choice and a core contribution of our work. A key insight of this paper is that the hidden states of large reasoning models naturally encode different aspects of the degree of reasoning sufficiency, and extracting signals from these representations allows the system to capture information about the reasoning process. We consider the discovery of signals inherently embedded in hidden states that reflect the degree of reasoning sufficiency, the analysis of how these signals behave throughout the reasoning trace, and the development of an end-to-end methodology and framework that leverages these signals from training to inference as one of the central contributions of this work. This multi-signal perspective is central to SpecExit and is a major reason why it generalizes across models and yields substantial end-to-end performance gains.

---

> ### Author Response · Authors · 2025-12-02
>
> Question1: We appreciate your concern. We would like to clarify that speculative decoding is theoretically lossless up to the limits of floating-point precision, as discussed in [1]. When applied correctly, speculative decoding does not degrade task accuracy or inference stability. Regarding the need for retraining, we would like to clarify that existing draft-model-based acceleration methods such as EAGLE and Medusa also require model-specific training. In this context, our Early-Exit Predictor can be trained jointly with the draft model at negligible cost or be further adapted on top of a trained draft model with 1-3 additional epochs. This design preserves the system’s lightweight nature and fits naturally within the broader landscape of speculative decoding accelerators.
>
> [1] Charlie Chen, Sebastian Borgeaud, Geoffrey Irving, Jean-Baptiste Lespiau, Laurent Sifre, and John Jumper. Accelerating large language model decoding with speculative sampling. arXiv preprint arXiv:2302.01318, 2023.
>
> Question2: We appreciate your request for empirical evidence on additional model families such as Phi and Mistral. In the revised manuscript, we have added experiments on Phi-4-reasoning and included the full set of results in **Table 1 of Section 4.2**. The added results further support the generalizability of the method beyond the models originally evaluated.
>
> For completeness, we reproduce the updated results table here:
>
> |           | GSM8K |      |       | MATH500 |      |       | AIME |       |       | HUMANEVAL+ |      |       | GPQA-D |      |        | ARC-Challenge |      |       |
> | --------- | ----- | ---- | ----- | ------- | ---- | ----- | ---- | ----- | ----- | ---------- | ---- | ----- | ------ | ---- | ------ | ------------- | ---- | ----- |
> | Method    | Acc   | Tok  | Lat   | Acc     | Tok  | Lat   | Acc  | Tok   | Lat   | Acc        | Tok  | Lat   | Acc    | Tok  | Lat    | Acc           | Tok  | Lat   |
> | Vanilla   | 95.8  | 709  | 207.2 | 94.9    | 2122 | 543.7 | 74.2 | 10980 | 536.3 | 72.6       | 2059 | 300.2 | 68.7   | 7544 | 726.7  | 96.6          | 607  | 193.2 |
> | NoThink   | 95.7  | 668  | 197.3 | 94.1    | 2051 | 554.8 | 70.7 | 11104 | 509.4 | 72.8       | 1919 | 297.3 | 64.7   | 7334 | 710.0  | 96.7          | 588  | 178.0 |
> | DEER      | 95.5  | 582  | 223.5 | 92.4    | 1502 | 516.0 | 60.0 | 7003  | 507.4 | 66.3       | 1420 | 211.1 | 65.2   | 4479 | 1296.1 | 96.0          | 540  | 183.4 |
> | EAGLE3    | 95.2  | 707  | 153.1 | 94.5    | 2136 | 324.2 | 74.0 | 10657 | 308.5 | 72.0       | 2035 | 155.8 | 68.2   | 7512 | 478.2  | 96.7          | 615  | 128.9 |
> | SpecExit* | 95.8  | 400  | 61.0  | 93.6    | 1750 | 271.9 | 74.7 | 9988  | 272.6 | 72.0       | 1605 | 131.8 | 67.7   | 6922 | 422.9  | 95.8          | 286  | 80.0  |

---

### Official Review · Reviewer_UuSi · 2025-10-31

**Soundness:** 2
**Presentation:** 3
**Contribution:** 2
**Rating:** 2
**Confidence:** 4

**Summary:**

The work introduces SpecExit, a reasoning-aware early exit framework for reducing the overall latency of the reasoning process. The method involves using a combination of speculative decoding and multi-task learning, to generate confidence signals from the draft model which guide the system to pre-empt the reasoning process eﬃciently. Evaluation across a range of tasks, the work claims to achieve 66% reduction in average generation length and 2.5x speedup in end-to-end latency, compared to the baseline.

**Strengths:**

1. The paper combines speculative decoding, multi-task learning, and confidence-based exit criteria in a coherent way to address reasoning efficiency.

2. Demonstrates significant inference latency reduction with minimal accuracy degradation across tasks.

3. The analysis of different signal types (Confidence / Progress / Remaining Reasoning Length) and smoothing methods (No Smoothing / Momentum / Sliding Window / EWMA) is thorough and informative.

4. The method offers a general framework applicable to existing reasoning models without modifying their core architecture.

**Weaknesses:**

- Line 267: The notion of “recover token” is unclear. How does it differ from the reasoning-end marker  ("\</think>")?
- Lines 363–364: The variant using combined signals (Confidence + Progress + Remaining Length) is denoted as SpecExit*, yet Table 1 reports results for SpecExit. Which configuration do these results correspond to?
- Table 1: Ambiguity exists between “Think” (line 333) and “Vanilla” (line 339). If they refer to the same baseline, HUMANEVAL+ accuracies (90.9 vs 88.4) are inconsistent.
- Table 1 caption should define what boldface indicates (best overall or best among comparable methods).

---
- Lines 313–314 mention that the speculative decoding baseline is trained on the same dataset as EAGLE3, but the draft model configuration is unspecified.
- What is the relationship between the draft and target models (same family, smaller variant, or identical)? Without this detail, reproducibility and interpretation of results are limited.

---
- Please quantify additional compute (e.g., FLOPs or GPU hours) for:
    - Dataset construction, which involves multiple reasoning traces per prompt to determine optimal lengths.
    - Multi-task learning with additional loss heads.
- These costs should be weighed against runtime improvements.
---

- Does the target LLM architecture affect training of the MLP heads? Since the dataset is derived from a specific model’s reasoning traces, how is overfitting avoided? Are exit heads transferable across models within the same family?
---

- The inference flow involving the MTP mechanism, lightweight draft model, MLP head, and target model is confusing. Please clarify whether:
    - The draft model includes MTP heads,
    - These predictions are passed to the target for verification, and
    - The target model then generates extended outputs via a linear layer.
- A schematic or algorithmic diagram would make the inference pipeline clearer.
---

- What is the motivation for three auxiliary losses (L_conf, L_prog, L_rem). The rationale for choosing these specific losses is unclear. What intuition or empirical observation led to defining these signals? Figure 5 shows dataset-dependent variation for Progress and Remaining signals, while Confidence remains consistently low. What happens if only Progress and Remaining signals are used?
- The abstract claims an “average generation length reduction of 66%,” but results suggest this is the maximum observed, not the mean. The phrasing should be corrected to “up to 66% reduction.”

**Questions:**

- Please provide detailed configurations of the draft model and its relation to the target model (architecture, size, tokenizer compatibility).

- Can the MLP exit heads trained on one model generalize to another of similar architecture?

- How large is the training overhead compared to runtime gains?

- Could the authors clarify whether the method assumes the draft and target models belong to the same family?

- Why were the three particular auxiliary losses (L_conf, L_prog, L_rem) chosen, and is there theoretical justification for their complementarity?

---

> ### Author Response · Authors · 2025-11-19
> **Response to Reviewer UuSi (Part 1/5)**
>
> We thank the reviewer for the insightful and valuable comments. We respond to each comment as follows and sincerely hope that our responses could properly address your concerns. We have updated our manuscript and highlighted the revisions for ease of inspection.
>
>
> The raised comments primarily concern the **clarification of basic concepts** (Weakness 1.1, 2, 4, 5; Question 1, 2, 4), the **precision of wording and terminology** (Weakness 1.2, 1.3, 1.4, 6.2), the **quantification of training costs** (Weakness 3; Question 3), and the **design rationale for the losses and signals** (Weakness 6; Question 5). We address each category of comments in detail below.
>
>
> > Weakness 1.1: Line 267: The notion of “recover token” is unclear. How does it differ from the reasoning-end marker ("`</think>`")?
>
> Thank you for pointing out the unclear definition of the “recover token,” which prompted us to introduce additional foundational knowledge on speculative decoding.
>
> In the speculative decoding inference pipeline, each sequence of draft tokens proposed by the draft model is forwarded to the target model for verification. Only those draft tokens that pass this verification are directly committed to the KV-cache of the target model, and the last accepted draft token is followed by a recover token generated by the target model. The hidden states produced by the target model during verification are then fed back into the draft model to guide the generation of new draft tokens. Subsequently, both the recover token and the newly generated draft tokens are again passed to the target model for verification and potential acceptance.
>
>  As mentioned above, the recover token is a replacement token proposed by the target model after rejecting a draft token. The reasoning-end marker (e.g., `</think>`) is a special token used to indicate the semantic termination of the reasoning process. Replacing the recover token with the reasoning-end marker has several advantages:
>
> We observed that directly replacing the paragraph delimiters (such as `.\n\n`) with `</think>` degrades the model’s instruction-following ability because the sentence becomes syntactically incomplete. Therefore, we choose to insert `</think>` after `.\n\n`. Speculative decoding naturally provides a mechanism for this, as described in Algorithm 1 of our paper: the target model produces a recover token after the last accepted token, and accepted tokens are directly appended to the KV-cache. We modify the accept length to stop at the `.\n\n` boundary, and replace the recover token with `</think>` as the next input. This allows us to complete early-exit `</think>` insertion in a single forward pass without additional cache manipulation, making the engineering implementation significantly simpler.
>
> The speculative decoding inference pipeline, including the definition and role of the recover token, has been added to **Section 3.3**.
>
>
>
> > Weakness 1.2: Lines 363–364: The variant using combined signals (Confidence + Progress + Remaining Length) is denoted as SpecExit*, yet Table 1 reports results for SpecExit. Which configuration do these results correspond to?
>
> We appreciate your careful observation. The results in Table 1 correspond to the SpecExit variant, where all tunable parameters are set to their default configuration. To avoid ambiguity, we have revised the table caption and consistently use “SpecExit*” in Table1. The updated caption explicitly clarifies that SpecExit* denotes the default configuration consistent with the best-performing variant in the ablation studies.
>
>
>
>
> > Weakness 1.3: Table 1: Ambiguity exists between “Think” (line 333) and “Vanilla” (line 339). If they refer to the same baseline, HUMANEVAL+ accuracies (90.9 vs 88.4) are inconsistent.
>
> Thank you for pointing this out. “Think” and “Vanilla” actually refer to the same baseline configuration, which serves as the full reasoning process without early exit. We have corrected the naming inconsistency by unifying the terminology to **“Vanilla”** throughout the paper.
> And for clarification, **NoThink** denotes the contrastive baseline where a closing `</think>` token is inserted at the beginning of the reasoning sequence. As described in line 353 in our submitted pdf (“Notably, for Qwen34B-Thinking-2507, inserting the `</think>` token in the NoThink baseline still fails to suppress reasoning, resulting in even longer outputs than the original Think mode.”).
>
>
> > Weakness 1.4: Table 1 caption should define what boldface indicates (best overall or best among comparable methods.
>
> We appreciate this suggestion. The revised caption now explicitly states the meaning of formatting styles: for early-exit methods (NoThink, DEER, SpecExit*), the highest and
> second-highest Acc values are highlighted in bold and \underline{underline}, respectively, while across all methods, the smallest and second-smallest Tok and Lat values. This clarification has been incorporated in the updated Table 1 caption.

---

> ### Author Response · Authors · 2025-11-19
> **Response to Reviewer UuSi (Part 2/5)**
>
> > Weakness 2.1: Lines 313–314 mention that the speculative decoding baseline is trained on the same dataset as EAGLE3, but the draft model configuration is unspecified.
>
> > Weaknesses2.2: What is the relationship between the draft and target models (same family, smaller variant, or identical)? Without this detail, reproducibility and interpretation of results are limited.
>
> > Question 1: Please provide detailed configurations of the draft model and its relation to the target model (architecture, size, tokenizer compatibility)
>
> > Question 4: Could the authors clarify whether the method assumes the draft and target models belong to the same family?
>
> We appreciate your observation regarding the lack of clarity in the description of the speculative decoding baseline and the relationship between the draft and target models. We have revised the manuscript accordingly and added the description of the draft model setup in the experimental section.
>
> To ensure a fair comparison with EAGLE3 in Table 1 and to facilitate a consistent implementation within the vLLM inference framework, our SpecExit system is trained together with the EAGLE3 draft model. Specifically, the draft model adopts the exact architecture used in EAGLE3: a one-layer causal language model (`Eagle3LlamaForCausalLM`) whose hidden size is matched to its corresponding target model.
>
> We have now included the detailed draft–target configurations, summarized in the following table:
>
> | Model name                   | Role (Target/Draft)    | Architecture     | Hidden_size | Layers | Vocab  |
> | ---------------------------- | ---------------------- | ---------------- | ----------- | ------ | ------ |
> | Qwen3-4B-Thinking-2507       | Target                 | Qwen3ForCausalLM | 2560        | 36     | 151936 |
> |        | Draft                        | Eagle3LlamaForCausalLM | 2560             | 1           | 32000  |
> | DeepSeek-R1-Distill-Llama-8B | Target                 | LlamaForCausalLM | 4096        | 32     | 128256 |
> |        | Draft                        | Eagle3LlamaForCausalLM | 4096             | 1           | 32000  |
>
> For each pair, the Eagle3 draft model shares the *same input embedding layer* as the target model (reusing the `model.embed_tokens` weights), ensuring full tokenizer and embedding compatibility. The draft model maintains its own output head and a compact vocabulary of 32k high-frequency tokens.
>
> We have added detailed descriptions of the experimental setup in **Section 4.1** and the model architectures in **Appendix A.3**. We hope the updated description clarifies these aspects more effectively.

---

> ### Author Response · Authors · 2025-11-19
> **Response to Reviewer UuSi (Part 3/5)**
>
> > Weakness 3: Please quantify additional compute (e.g., FLOPs or GPU hours) for:
> > - Dataset construction, which involves multiple reasoning traces per prompt to determine optimal lengths.
> > - Multi-task learning with additional loss heads.
> >
> > These costs should be weighed against runtime improvements.
>
> > Question 3: How large is the training overhead compared to runtime gains?
>
> We appreciate your insightful question regarding the additional compute required by (i) dataset construction through multiple reasoning traces and (ii) multi-task learning with auxiliary loss heads, as well as how these costs compare to runtime gains. We provide a detailed quantification below.
>
> First, we emphasize that we operate within a single draft model (EAGLE3 draft model), which itself must undergo additional training.Our early-exit supervision is added on top of this existing speculative decoding training. Importantly, the extra supervision can be trained jointly with the EAGLE3 objective at low marginal cost.
>
> **Dataset construction**: To determine optimal segmentation points for early-exit supervision, we evaluate each existing reasoning trace at up to five candidate paragraph-boundary positions (based on evenly dividing all split indices). At each candidate position, we insert a `</think>` token, decode up to 500 tokens, and verify whether the predicted answer is correct. Thus, one sequence triggers at most five forward passes, generating up to 2,500 tokens in total.
> We measured this overhead using 5,000 randomly sampled examples from the MATH training set and executed the process through parallel vLLM service calls on an 8×H20 cluster. The complete data construction phase required **0.6h (4.8 H20 GPU Hours)**.
>
> **Signals Training**: The auxiliary exit-signal heads add only a small linear projection applied to the hidden state. Let's denote the hidden state as `x` with shape `(s, d)` (sequence length `s`, hidden dimension `d`), and the linear projection weight as `W` with shape `(d, 3)`. The output `y` has a shape of `(s, 3)`.
>
> Forward Pass: The total cost includes the linear projection and the loss calculation.
> 1.  Linear Projection: The primary computation is the matrix multiplication `y = x @ W`.
>     *   FLOPs = 2 * s * d * 3 = 6sd.
> 2.  Loss Calculation: This involves element-wise operations (Sigmoid, MSE, MSLE).
>     *   FLOPs = s * (FLOPS(MSE)*2 + FLOPS(sigmoid) + FLOPS(MSLE)) ≈ 15s. This cost is linear in sequence length `s`.
>
> Backward Pass: The backward pass computes gradients for the weights (`dL/dW`) and the input (`dL/dx`).
> 1.  Loss Gradient: Backpropagating through the loss functions to get `dL/dy` also involves element-wise operations.
>     *   FLOPs ≈ O(s).
> 2.  Weight & Input Gradients: These are dominated by matrix multiplications.
>     *   `dL/dW = x.T @ (dL/dy)`:  FLOPs ≈ 2 * d * s * 3 = 6sd.
>     *   `dL/dx = (dL/dy) @ W.T`:  FLOPs ≈ 2 * s * 3 * d = 6sd.
>
> Thus, the **Total FLOPs ≈ 18sd + O(s)**. Therefore, this linear projection head constitutes a tiny fraction compared to the entire draft model's parameters.
>
>
> Training a full epoch of our SpecExit modules (EAGLE3 + early-exit module) requires approximately **1.5 h (12 H20 GPU hours)**. The exit-signal heads typically converge within **1–3 epochs**, and because their regression objectives are orthogonal to the token-prediction objective, these heads can be trained jointly with the speculative decoding model or added through a short, low-cost continued training phase.
>
> Both sources of overhead are therefore small. By contrast, our method achieves **up to 66% reduction in reasoning length** and **2.5× end-to-end inference speedup** compared with EAGLE3 baseline. Consequently, the modest training overhead is far outweighed by the substantial runtime benefits, and the method can be integrated into existing speculative decoding training pipelines with minimal additional cost.

---

> ### Author Response · Authors · 2025-11-19
> **Response to Reviewer UuSi (Part 4/5)**
>
> > Weakness 4: Does the target LLM architecture affect training of the MLP heads? Since the dataset is derived from a specific model’s reasoning traces, how is overfitting avoided? Are exit heads transferable across models within the same family?
>
> > Question 2: Can the MLP exit heads trained on one model generalize to another of similar architecture?
>
> We appreciate your insightful questions.
>
> First, our SpecExit mechanism exclusively extracts information from intermediate hidden states and does not rely on any specific architectural features of the target model. Therefore, the training procedure of the exit heads is largely architecture-agnostic.
>
> Second, to mitigate overfitting problem, we leverage the dynamic weighting strategy described in **Section 3.2** of the paper. Since the regression losses for the exit signals converge faster than the token classification loss, the gradient-based dynamic weighting mechanism balances the contributions of different tasks. Formally, the total loss is computed as:
>
>
> $$
> \mathcal{L}\_{\text{total}} = \mathcal{L}\_{\text{cls}} + \sum\_j \lambda\_j \mathcal{L}\_j,
> \quad
> \lambda\_j = \frac{|\nabla\_\theta \mathcal{L}\_j|}{\sum\_{k} |\nabla\_\theta \mathcal{L}\_k|},
> $$
>
> where $\mathcal{L}\_j \in {\mathcal{L}\_{\text{conf}}, \mathcal{L}\_{\text{prog}}, \mathcal{L}\_{\text{rem}}}$.
>
>  By adaptively reweighting tasks according to their gradient magnitudes, the dynamic weighting mechanism continuously adjusts the relative importance of each objective. In particular, in cases such as those illustrated in Figure 3 where the regression losses of the exit signals converge substantially faster than the token-classification loss, this strategy effectively mitigates overfitting of the exit-signal regression heads.
>
> Finally, regarding transferability, the exit heads cannot be directly reused across models, even within the same family, because the hidden state shapes differ between models. Regarding the need for retraining, we would like to clarify that existing draft-model-based acceleration methods such as EAGLE and Medusa also require model-specific training. In this context, our Early-Exit Predictor can be trained jointly with the draft model at negligible cost or be further adapted on top of a trained draft model with 1-3 additional epochs. This design preserves the system’s lightweight nature and fits naturally within the broader landscape of speculative decoding accelerators.
>
>
>
> > Weakness 5: The inference flow involving the MTP mechanism, lightweight draft model, MLP head, and target model is confusing. Please clarify whether: The draft model includes MTP heads, These predictions are passed to the target for verification, and The target model then generates extended outputs via a linear layer. A schematic or algorithmic diagram would make the inference pipeline clearer.
>
> Thank you for highlighting the need to further clarify the inference procedure involving the multi-token prediction (MTP) mechanism, the lightweight draft model, the MLP head, and the target model. We have revised the manuscript to improve clarity. Specifically, we added a more detailed explanation in Section 3.3, supplemented by the schematic in Figure 4 and the accompanying Algorithm 1, which together provide a step-by-step illustration of the full inference pipeline. In addition, we included a more comprehensive diagram in **Figure 11 in Appendix A.3**, offering a higher-level visualization of the integrated inference pipeline.
>
> In our framework, the draft model produces draft tokens conditioned on previously accepted tokens and on the hidden states supplied by the target model. These draft tokens are passed to the target model for verification. During the verification step, the target model computes logits for token acceptance and generates hidden states, which are subsequently processed by a lightweight MLP head to obtain three early-exit signals: confidence, reasoning progress, and predicted remaining reasoning length. Accepted draft tokens are appended to the target model’s KV-cache, after which the target model generates a recover token following the last accepted token. The hidden states obtained in this verification step are then fed back into the draft model to guide the next set of draft tokens. This iterative interaction between the draft model and the target model continues until the stopping criterion is triggered, as illustrated in Figure 4.
>
> We appreciate your feedback, which helped us clarify the description of our inference flow and improve the presentation of the method.

---

> ### Author Response · Authors · 2025-11-19
> **Response to Reviewer UuSi (Part 5/5)**
>
> > Weakness 6.1: What is the motivation for three auxiliary losses (L_conf, L_prog, L_rem). The rationale for choosing these specific losses is unclear. What intuition or empirical observation led to defining these signals? Figure 5 shows dataset-dependent variation for Progress and Remaining signals, while Confidence remains consistently low. What happens if only Progress and Remaining signals are used?
>
> > Questions 5: Why were the three particular auxiliary losses (L_conf, L_prog, L_rem) chosen, and is there theoretical justification for their complementarity?
>
> Thank you for the insightful question. The review appears to conflate two distinct conceptual points: the **design of the loss functions** and the **selection of the signal types**. To clarify, our response will address these two aspects separately.
>
> (1) **The design of the loss functions**:
>
> The formulations of the losses $(L_{\text{conf}}, L_{\text{prog}}, L_{\text{rem}})$ primarily follow from both the intrinsic numerical characteristics of each signal and empirical observations of hidden-state dynamics.
>
> First, each loss is matched to the natural scale of its target signal. The confidence and progress signals lie in ([0,1]), so we apply a sigmoid activation and regress using a mean squared error objective:
> $$
> \mathcal{L}\_{\text{conf}}=\tfrac{1}{N}\sum\_i (\sigma(\hat c\_i)-c\_i)^2,\quad
> \mathcal{L}\_{\text{prog}}=\tfrac{1}{N}\sum\_i (\sigma(\hat p\_i)-p\_i)^2.
> $$
>
> In contrast, the remaining-length signal ranges from 0 to over 30,000 tokens, so we apply a logarithmic transformation to avoid scale imbalance:
> $$
> \mathcal{L}\_{\text{rem}}=\tfrac{1}{N}\sum\_i (\hat r\_i-\log(1+r\_i))^2.
> $$
>
> (2) **The selection of the signal types**:
>
> The choice of these three signals is further supported by empirical analyses of hidden-state representations, following your suggestion. As reported in **Figure 7(a) in Appendix A.2**, PCA and UMAP reveal clear clustering patterns corresponding to early, intermediate, and late reasoning positions across datasets of varying difficulty. As shown in **Figure 7(b) in Appendix A.2**, hidden states also form meaningful clusters when grouped by geometric-mean paragraph probabilities, indicating that confidence-related information emerges naturally in the representation space.
>
> We additionally analyzed the statistical behavior of the three predicted signals. **Figure 9 in Appendix A.2** shows that their distributions at stopping positions differ significantly between correct and incorrect answers. Qualitative case studies in **Figures 13 and 14 in Appendix A.4** further illustrate that the three signals capture distinct and complementary aspects of reasoning sufficiency.
>
> As shown in **Figure 8 in Appendix A.2**, compared with the *Progress–Remain-Only* setting, using all three signals achieves a more favorable trade-off between maintaining accuracy and reducing the reasoning length.
>
> In addition, we provide methods for determining thresholds through statistical analyses or threshold-search procedures, with the corresponding experiments included in **Figures 9 and 10 in Appendix A.2**. These methods ensure that if one signal becomes relatively less informative, its threshold naturally falls back to a more permissive constraint, thereby improving the generality and robustness of our approach.
>
> Together, these analyses provide both empirical motivation and preliminary theoretical justification for including all three auxiliary losses and signals. We have incorporated the expanded motivation, ablations, and discussions into the appendix for clarity.
>
>
>
>
> > Weakness 6.2: The abstract claims an “average generation length reduction of 66%,” but results suggest this is the maximum observed, not the mean. The phrasing should be corrected to “up to 66% reduction.”
>
> Thank you for pointing it out. What you understand is correct. The "average" term means the generation length of that task is averaged, but it may have caused confusion with the overall mean across all tasks. We have corrected the phrasing to avoid misunderstanding.

---

### Official Review · Reviewer_rpsy · 2025-11-01

**Soundness:** 3
**Presentation:** 2
**Contribution:** 3
**Rating:** 6
**Confidence:** 3

**Summary:**

This paper introduces SpecExit, a novel framework designed to accelerate inference in Large Reasoning Models (LRMs) by tackling the "overthinking" problem, where models produce unnecessarily long reasoning chains, leading to high latency. The core idea is to integrate an early-exit mechanism directly into the speculative decoding process. Unlike prior methods that rely on costly probing of the target model, SpecExit extends the lightweight draft model to predict not only future tokens but also three auxiliary signals: confidence, reasoning progress, and remaining reasoning length. These signals are learned via multi-task training on data where the minimal sufficient reasoning length is heuristically identified. During inference, these signals are used to dynamically terminate the reasoning process at natural breakpoints (e.g., paragraph ends) without any modification to the target model or extra probing overhead. Experiments on various reasoning benchmarks with models like Qwen3-4B and DeepSeek-R1-Distill show that SpecExit can reduce generation length by up to 66% and achieve end-to-end latency speedups of up to 2.5x compared to a standard speculative decoding baseline, with minimal impact on task accuracy.

**Strengths:**

The primary strength of this work lies in its novel and elegant integration of two distinct lines of research: speculative decoding for per-token acceleration and early-exit mechanisms for reducing reasoning length. Instead of treating these as separate problems, the authors propose a unified framework. The key insight—to offload the exit-signal prediction to the draft model and leverage its hidden states—is highly original. This avoids the well-documented overhead of probing-based early-exit methods, which is a creative and impactful conceptual leap.

**Weaknesses:**

1. The author should introduce some basic knowledge of speculative decoding to facilitate readers who are not familiar with speculative decoding to understand the content of the paper.

2. Figure 2 only shows the forward process of one token. I don't understand how to predict the vocab logits and signals of multiple tokens simultaneously. The author needs to explain in detail how the MTP layer in the paper is designed. Is it exactly the same structure as Medusa or EAGLE?

3. The author combines speculative decoding with exit prediction, but I believe there is no coupling relationship between the two. According to the method designed in this paper, adding an MLP layer directly to the hidden feature of the token can also predict the exit signal without the need for an MTP layer.

4. The author can supplement the experimental results without speculative decoding (only predicting the exit signal of each token).

**Questions:**

Please refer to Weaknesses.

---

> ### Author Response · Authors · 2025-11-19
> **Response to Reviewer rpsy (Part 1/2)**
>
> We thank the reviewer for the insightful and valuable comments. We respond to each comment as follows and sincerely hope that our responses could properly address your concerns. We have updated our manuscript and highlighted the revisions for ease of inspection.
>
> > Weakness 1: The author should introduce some basic knowledge of speculative decoding to facilitate readers who are not familiar with speculative decoding to understand the content of the paper.
>
> Thank you for this helpful suggestion. We have expanded the introduction section to provide more explanation of speculative decoding (SD) for readers less familiar with this technique. SD is introduced in our 3rd paragraph of introduction (line 46) and discussed in our 2nd paragraph of related work (line 465) in our submitted paper. It was proposed by [1][2], and recent state-of-the-art works include [3][4], all of which are cited in our paper. Additionally, in Section 3.3, we provide a more detailed explanation of the SD inference pipeline and its core concepts, including how draft tokens are proposed, verified, and committed, which naturally leads to Figure 4 and Algorithm 1 for a clearer visualization of the process.
>
> Here is a brief recap: standard autoregressive decoding generates tokens sequentially, which is memory-bound and underutilizes the parallel processing capabilities of GPUs. SD addresses this limitation by using a lightweight draft model to propose multiple candidate tokens, which are then verified in parallel by the target model in a single forward pass. This enables generating multiple tokens per forward round, thereby accelerating decoding speed.
> In SpecExit, we propose to apply SD in long chain-of-thought (CoT) reasoning tasks, where decoding time dominates the end-to-end latency. Accelerating the decoding phase thus directly reduces the overall latency of reasoning tasks.
>
> [1] Chen C, Borgeaud S, Irving G, et al. Accelerating large language model decoding with speculative sampling. arXiv preprint arXiv:2302.01318, 2023.
>
> [2] Leviathan Y, Kalman M, Matias Y. Fast inference from transformers via speculative decoding. International Conference on Machine Learning, 2023.
>
> [3] Li Y, Wei F, Zhang C, et al. EAGLE: Speculative sampling requires rethinking feature uncertainty. International Conference on Machine Learning, 2024.
>
> [4] Zhang L, Wang X, Huang Y, et al. Learning harmonized representations for speculative sampling. International Conference on Learning Representations, 2025.
>
>
>
> > Weakness 2: Figure 2 only shows the forward process of one token. I don't understand how to predict the vocab logits and signals of multiple tokens simultaneously. The author needs to explain in detail how the MTP layer in the paper is designed. Is it exactly the same structure as Medusa or EAGLE?
>
> Thank you for the careful reading and insightful comments. Our SpecExit early-exiting mechanism is orthogonal to the underlying multi-token prediction design and can be combined with EAGLE3/Medusa. In our experiments, to ensure a fair comparison with EAGLE3, we integrate our SpecExit mechanism into the EAGLE3 framework.
>
> To jointly predict vocab logits and the three auxiliary signals within a single forward pass, we fuse the Linear layer of the MTP head. Instead of using a projection that outputs only vocab logits, the extended projection simultaneously produces (i) the logits the logits that are sent to the LM Head and (ii) the three scalar signals for each token. This fused design allows the hidden state of each token to be transformed once, after which the output is split into the standard vocabulary logits and the three signals required by SpecExit. Thus, the prediction for all tokens and all signals is computed in parallel without introducing additional forward passes or architectural branches.

---

> ### Author Response · Authors · 2025-11-19
> **Response to Reviewer rpsy (Part 2/2)**
>
> > Weakness 3: The author combines speculative decoding with exit prediction, but I believe there is no coupling relationship between the two. According to the method designed in this paper, adding an MLP layer directly to the hidden feature of the token can also predict the exit signal without the need for an MTP layer.
>
> > Weakness 4: The author can supplement the experimental results without speculative decoding (only predicting the exit signal of each token).
>
> We greatly appreciate your keen insight into this aspect of our design. Indeed, in terms of parameterization, the early-exit predictor is orthogonal to speculative decoding, and the exit mechanism can operate independently. Our joint design provides  benefits as follows:
>
> (1) By training the speculative draft model and the early-exit mechanism simultaneously, we can achieve superior acceleration within a single training process. Furthermore, the module used for early-exit prediction can be fused with the FC Layer used in speculative decoding, enabling us to reuse hidden features and thereby achieve additional speed-ups beyond pure early exit or pure speculative decoding alone.
>
> (2)The integration allows us to implement one-forward insertion of the `</think>` token. We observed that directly replacing the paragraph delimiters (such as `.\n\n`) with `</think>` degrades the model’s instruction-following ability because the sentence becomes syntactically incomplete, so we choose to insert `</think>` after `.\n\n`. Speculative decoding naturally provides a mechanism for this, as described in Algorithm 1 of our paper: the target model produces a recover token after the last accepted token, and accepted tokens are directly appended to the KV-cache. We modify the accept length to stop at the `.\n\n` boundary, and replace the recover token with `</think>` as the next input. This allows us to complete early-exit `</think>` insertion in a single forward pass without additional cache manipulation, making the engineering implementation significantly simpler.
>
> In response to your insightful comment, we conducted additional experiments evaluating the exit-only setting without speculative decoding. As vLLM introduces non-trivial engineering workload for injecting an additional standalone MLP without speculative decoding, we report PyTorch-based results here. The results are shown below:
>
> | Dataset   | MATH500 |      |       | GSM8K |      |      | GPQA |      |       |
> | --------- | ------- | ---- | ----- | ----- | ---- | ---- | ---- | ---- | ----- |
> | Method    | Acc     | Tok  | Lat   | Acc   | Tok  | Lat  | Acc  | Tok  | Lat   |
> | Vanilla   | 96.4    | 6661 | 169.8 | 95.0  | 1518 | 38.9 | 68.2 | 6672 | 170.8 |
> | Exit-Only | 95.8    | 5242 | 134.2 | 93.8  | 538  | 13.6 | 69.2 | 8998 | 229.7 |

---

### Official Review · Reviewer_4V7z · 2025-11-02

**Soundness:** 3
**Presentation:** 3
**Contribution:** 3
**Rating:** 6
**Confidence:** 4

**Summary:**

This paper introduces **SpecExit**, a *reasoning-aware early-exit framework* designed to reduce overthinking in large reasoning models (LRMs).
Instead of relying on explicit probing phrases (e.g., “Final Answer is”), SpecExit leverages **hidden states** from a lightweight **draft model** to jointly predict (i) future tokens and (ii) early-exit signals—**confidence**, **reasoning progress**, and **remaining reasoning length**—in a single forward pass.
The method removes probing overhead, integrates smoothly with **speculative decoding**, and dynamically stops generation at semantically coherent boundaries such as paragraph delimiters.
Experiments on GSM8K, MATH500, AIME, GPQA, HumanEval+, and ARC-Challenge show up to **66 % shorter reasoning chains** and **2.5 × latency reduction** without accuracy loss.
Ablation studies confirm that multi-signal integration with EWMA smoothing yields the best balance between efficiency and correctness.

**Strengths:**

* Novel integration of early-exit and speculative decoding via hidden-state supervision.
* Eliminates probing overhead while maintaining accuracy.
* Multi-signal design (confidence + progress + remaining length) improves robustness across tasks.
* Experiments on six benchmarks and two reasoning-optimized backbones.
* Ablations on signal types, smoothing strategies, and segmentation markers.
* Reproducibility and open-source release fully documented.
* Clear contextualization within reasoning-efficiency literature (DEER, EAGLE, RL-based control).

**Weaknesses:**

* Lacks a **formal theoretical analysis** of why hidden-state features correlate with reasoning sufficiency.
* Dependence on explicit *<think>* / *</think>* markers may limit generalization to free-form reasoning.
* Limited model family evaluation; Experiments are conducted on two reasoning model families.

**Questions:**

1. How sensitive is SpecExit to the chosen thresholds (confidence > 0.8, progress > 0.3)? Could these be learned dynamically?
2. Have you tested **cross-model transferability** of exit heads (e.g., trained on Qwen, applied to DeepSeek)?
3. How would SpecExit behave with **long-context**?
4. Could you quantify **energy / GPU-hour savings** corresponding to the reported 2.5× latency reduction?
5. Can you provide a theoretical account—stating assumptions and guarantees—that explains why functions of intermediate hidden states serve as reliable proxies for ‘reasoning sufficiency’ (e.g., monotonicity/calibration of the confidence and progress heads)?

---

> ### Author Response · Authors · 2025-11-19
> **Response to Reviewer 4V7z (Part 1/2)**
>
> We thank the reviewer for the insightful and valuable comments. We respond to each comment as follows and sincerely hope that our responses could properly address your concerns. We have updated our manuscript and highlighted the revisions for ease of inspection.
>
> >  Question 1: How sensitive is SpecExit to the chosen thresholds (confidence > 0.8, progress > 0.3)? Could these be learned dynamically?
>
> We sincerely appreciate your sharp observation regarding the sensitivity of SpecExit to threshold choices and the possibility of learning such thresholds dynamically. To address this concern comprehensively, we have added further analysis in **Appendix A.2**, where we provide two approaches for learning adaptive thresholds.
>
> Specifically, we sample 90 instances of varying difficulty from the validation split of training data and design two methods of threshold calibration procedures based on this held-out set.
>
> (1) We conduct a statistical distribution–based threshold inference. During data construction, we have access to the shortest valid reasoning path for each problem, allowing us to determine whether stopping at any paragraph boundary would still lead to a correct answer. By examining the empirical distributions of confidence, progress, and remaining-length signals with correctness, we obtain the distributions shown in **Figure 9 in Appendix A.2**. Therefore, we can estimate thresholds by analyzing the empirical value distributions of each signal under correct and wrong stopping decisions, and selecting threshold based on their separation in these distributions.
>
> (2) We perform a systematic design space exploration (DSE) over the three-dimensional threshold space defined by confidence, progress, and remaining-length values. Leveraging a vLLM-based evaluation pipeline, we conduct a full grid search on the 90-sample calibration set, requiring approximately 2.5 hours on an 8×H20 cluster. From all threshold combinations, we select those on the Pareto frontier that achieve the best trade-off between accuracy preservation and reduction in reasoning length. As shown in **Figure 10 in Appendix A.2**, SpecExit remains robust across a broad threshold range, while the grid search further reveals fine-grained performance–efficiency trade-offs beyond manual tuning.
>
> We have included these results and discussion in the **Appendix A.2** to directly address your insightful comments.
>
>
>
> > Question 2: QuestionHave you tested **cross-model transferability** of exit heads (e.g., trained on Qwen, applied to DeepSeek)?
>
> Thank you for raising this important question regarding the cross-model transferability of exit heads. We have not evaluated transferring exit heads across heterogeneous LLM families (e.g., training on Qwen and applying to DeepSeek), primarily because the exit mechanism in our framework is tightly coupled to the internal representation space of the underlying model.
>
> Our exit heads operate directly on the hidden states and learn to infer confidence, progress, and remaining steps from their distribution via small multilayer perceptrons. These hidden states, however, are not only model-specific in dimensionality (i.e., hidden size and layer structure) but also differ substantially in their representation space across model families due to differences in training corpora, tokenizer design, positional encoding, attention implementation, and normalization strategy. As a result, both the shape and the distributional characteristics of hidden representations vary sufficiently that an exit head trained on one model family cannot be directly applied to another without significant degradation.

---

> ### Author Response · Authors · 2025-11-19
> **Response to Reviewer 4V7z (Part 2/2)**
>
> > Question 3: How would SpecExit behave with **long-context**?
>
> Thank you for this insightful question. Our design motivation for SpecExit is to predict the shortest effective reasoning content and thus accelerate inference, especially for short-input, long-reasoning tasks with relatively high difficulty.
>
> Regarding long-context scenarios, we refer to the results reported in the Qwen3 Technical Report (Appendix A.1.1 *Long-Context Ability*), where the RULER benchmark is evaluated as follows:
>
> | RULER                      | Avg. | 4K   | 8K   | 16K  | 32K  | 64K  | 128K |
> | :------------------------- | :--- | :--- | :--- | :--- | :--- | :--- | :--- |
> | Non-thinking Mode Qwen3-4B | 85.2 | 95.1 | 93.6 | 91.0 | 87.8 | 77.8 | 66.0 |
> | Thinking Mode Qwen3-4B     | 83.5 | 92.7 | 88.7 | 86.5 | 83.2 | 83.0 | 67.2 |
>
> As shown, non-thinking mode actually achieves better accuracy than thinking mode on RULER across most context lengths. In such a setting, using thinking mode plus SpecExit to truncate the reasoning would not provide a clear advantage over directly using non-thinking mode: the latter is already both simpler and more accurate.
>
> For this reason, we did **not** further explore SpecExit in long-context benchmarks like RULER in this work. Our current focus remains on short-input, long-output tasks where (i) thinking mode clearly improves quality over non-thinking mode, and (ii) there is substantial potential to save computation by exiting early. Studying how to best integrate SpecExit with long-context usage is an interesting direction for future work, but it is outside the scope of the present paper.
>
>
>
>
> > Question 4: Could you quantify **energy / GPU-hour savings** corresponding to the reported 2.5× latency reduction?
>
>
> Thank you for the suggestion. Below we provide the GPU-hour measurements and corresponding speedups of DeepSeek-R1-Distill-Llama-8B across different datasets.
>
> |            | GSM8K |          |         | MATH500 |          |         | AIME |          |         | HUMANEVAL+ |          |         | GPQA-D |          |         | ARC-Challenge |          |         |
> | ---------- | ----- | -------- | ------- | ------- | -------- | ------- | ---- | -------- | ------- | ---------- | -------- | ------- | ------ | -------- | ------- | ------------- | -------- | ------- |
> | **Method** | Acc   | GPU-Hour | Speedup | Acc     | GPU-Hour | Speedup | Acc  | GPU-Hour | Speedup | Acc        | GPU-Hour | Speedup | Acc    | GPU-Hour | Speedup | Acc           | GPU-Hour | Speedup |
> | EAGLE3     | 79.3  | 276.9    |         | 80.8    | 593.6    |         | 30   | 228.1    |         | 78.7       | 346.5    |         | 43.9   | 420.1    |         | 59.2          | 496.4    |         |
> | SpecExit*  | 75.3  | 112.6    | 2.5     | 80.6    | 348.3    | 1.7     | 36.7 | 176      | 1.3     | 81.7       | 118.1    | 2.9     | 46     | 307.5    | 1.4     | 50.3          | 253.7    | 2.0     |
>
>
>
> >  Question 5: Can you provide a theoretical account—stating assumptions and guarantees—that explains why functions of intermediate hidden states serve as reliable proxies for ‘reasoning sufficiency’ (e.g., monotonicity/calibration of the confidence and progress heads)?
>
> Thank your for the insightful comment regarding the theoretical justification of using intermediate hidden states as proxies for reasoning sufficiency. Following your suggestion, we conducted supplementary analyses of hidden-state representations. The results are reported in **Figure 7 in Appendix A.2**. By applying Principal Component Analysis (PCA) and Uniform Manifold Approximation and Projection (UMAP), we observed clustering patterns that correspond to early, intermediate, and final reasoning positions across datasets of varying difficulty. Hidden states also form meaningful clusters when grouped by the geometric mean of paragraph-level probabilities, which reflects confidence levels.
>
> In addition, we examined the statistical distributions and concrete case studies of our predicted signals. **Figure 9 in Appendix A.2** shows the distribution of signal values at the stopping positions in calibration data, revealing different distributions depending on whether the final answer is correct or incorrect. **Figures 13 and 14 in Appendix A.4** provide qualitative case analyses for simple and complex problems, respectively, demonstrating that the proposed multiple signals capture complementary aspects of reasoning sufficiency.
>
> Together, these empirical observations provide preliminary evidence that intermediate hidden states encode information relevant to reasoning progress and sufficiency. While a formal theoretical proof remains an important direction for future work, we believe these analyses substantially strengthen the empirical motivation for using functions of intermediate hidden states as proxies. We have added the above analyses and figures to **Appendix** for reference.

---

### Author Response · Authors · 2025-11-26
**Global Response to All Reviewers**

We sincerely thank all reviewers for their valuable feedback. To address the concerns, we have updated our manuscript and **highlighted** the revisions for ease of inspection. The revisions mainly include:

1. **Methodology Clarity**: We add introduction & explanation of speculative decoding and clarify the inference pipeline with improved figures and details in **Section 1 & 3.3** and **Figure 11 in Appendix A.3**.
2. **Signal Insight**: We provide empirical analysis of why reasoning signals can be learned from hidden states through PCA+UMAP analysis of hidden states clusters in **Figure 7 in Appendix A.2**. We also illustrate statistical distribution analysis of predicted signals in **Figure 9 in Appendix A.2**, and qualitative case studies in **Figures 13 and 14 in Appendix A.4**. These analyses demonstrate that intermediate hidden states naturally encode information about reasoning sufficiency, validating our contribution of extracting early-exit signals from hidden features.
3. **Threshold Calibration**: We propose two data-driven methods for learning adaptive thresholds in **Appendix A.2**: (1) a statistical approach that determines thresholds by leveraging the separation in empirical signal distributions between correct and incorrect stopping decisions, and (2) a systematic design space exploration that performs a grid search to locate thresholds which trade off between accuracy preservation and reduction in reasoning length.
4. **Experimental Details**:
  - Include evaluation on **Phi-4-reasoning** to demonstrate generalization across additional model families in **Table 1 of Section 4.2**.
   - Clarify draft-target model relationships and configurations in **Table 3 in Appendix A.3**.
   - Illustrate the effect of EWMA in case study in **Figure 15 in Appendix A.4**.
   - Quantify additional training overhead (modest vs runtime gains).
5. **Presentation**: Improve terminology consistency and refined the clarity of result reporting.

These revisions have strengthened the paper's clarity, robustness, and reproducibility. We appreciate insightful comments from all reviewers that helped improve our work.

---

### Meta-Review · Area_Chair_wqbN · 2026-01-07

**Summary:**

The reviewers recognized the importance of addressing "overthinking" in Large Reasoning Models (LRMs) and praised the novel integration of early-exit mechanisms with speculative decoding. However, several critical concerns were raised:
1. Theoretical Grounding: Reviewers (4V7z, 8nc3) requested a more formal analysis of why intermediate hidden states serve as reliable proxies for "reasoning sufficiency."
2. Generalizability and Scaling: Concerns were raised about the model-specific training requirement for exit heads and the lack of empirical evidence across diverse model families like Phi or Mistral (8nc3, 4V7z, UuSi).
3. Sensitivity and Robustness: The stability of raw signals and the sensitivity of the method to specific thresholds (confidence, progress, etc.) were questioned (4V7z, 8nc3, UuSi).

The tasks this paper is interesting and important but this paper is on the borderline.

**Reviewer Concerns:**

Addressed or partially addressed:
1. Expanded Empirical Evidence: The authors added experiments on Phi-4-reasoning, demonstrating that SpecExit generalizes to other architectures with competitive performance gains.
2. Signal and Threshold Analysis: The authors introduced two calibration methods: a statistical distribution-based approach and a Pareto-frontier grid search.
3. Computational Transparency Authors quantified the training overhead, showing that dataset construction (4.8 GPU hours) and multi-task training (12 GPU hours) are negligible compared to the 2.5× inference speedup.

Outstanding Concerns:
1. Cross-Model Transferability: As the authors noted, exit heads are tightly coupled to the hidden state representation of the specific model. While they are inexpensive to train, they currently lack "plug-and-play" transferability across heterogeneous model families.
2. Long-Context Interaction: The interaction between SpecExit and long-context benchmarks remains an open area for future research, as the current focus is on short-input, long-reasoning tasks.

**Reviewer Scores:**

4V7z	6->6	 Addressed theoretical proxies and energy quantification.
rpsy	 6->6	Positive. Satisfied with SD fundamentals and MTP layer explanations.
UuSi	2->2/4	Authors clarified terminology and cost concerns.
8nc3	6->6	While maintaining the score, the reviewer acknowledged the implementation steps and losslessness of SD.

---

### Decision · Program_Chairs · 2026-01-26

Reject